# Linear Uncertainty Quantification of Graphical Model Inference

**Chenghua Guo**[1*], **Han Yu**[2*], **Jiaxin Liu**[3], **Chao Chen**[4], **Qi Li**[5], **Sihong Xie**[6†], **Xi Zhang**[1†]

[1]Key Laboratory of Trustworthy Distributed Computing and Service (MoE),
Beijing University of Posts and Telecommunications, China
[2]College of Artificial Intelligence, Beijing University of Posts and Telecommunications, China
[3]Department of Computer Science and Engineering, Lehigh University, USA
[4]School of Computer Science and Technology, Harbin Institute of Technology (Shenzhen), China
[5]Department of Computer Science, Iowa State University, USA
[6]AI Thrust, The Hong Kong University of Science and Technology (Guangzhou), China
{chenghuaguo, zhangx}@bupt.edu.cn, yuhan2021@bupt.edu.cn,
jilb17@lehigh.edu, cha01nbox@gmail.com, qli@iastate.edu,
xiesihong1@gmail.com

## Abstract

Uncertainty Quantification (UQ) is vital for decision makers as it offers insights into the potential reliability of data and model, enabling more informed and risk-aware decision-making. Graphical models, capable of representing data with complex dependencies, are widely used across domains. Existing sampling-based UQ methods are unbiased but cannot guarantee convergence and are time-consuming on large-scale graphs. There are fast UQ methods for graphical models with closed-form solutions and convergence guarantee but with uncertainty underestimation. We propose *LinUProp*, a UQ method that utilizes a novel linear propagation of uncertainty to model uncertainty among related nodes additively instead of multiplicatively, to offer linear scalability, guaranteed convergence, and closed-form solutions without underestimating uncertainty. Theoretically, we decompose the expected prediction error of the graphical model and prove that the uncertainty computed by *LinUProp* is the *generalized variance component* of the decomposition. Experimentally, we demonstrate that *LinUProp* is consistent with the sampling-based method but with linear scalability and fast convergence. Moreover, *LinUProp* outperforms competitors in uncertainty-based active learning on four real-world graph datasets, achieving higher accuracy with a lower labeling budget.

## 1 Introduction

Graphical models are known for their capability to represent data with complex dependencies. These models have been extensively applied to various fields, ranging from social networks [9, 39], fraud detection [26, 8], recommendation [14, 37] and crowdsourcing [25, 16], to more recent applications in enhancing graph neural networks (GNNs) [10, 28] and large language models (LLMs) [15, 34]. Among the many inference techniques in graphical models [19, 13, 12, 17], Belief Propagation (BP) [27] stands out as a powerful iterative message-passing algorithm. BP takes an initial guess (or "prior belief") of each node and refines it using information propagation, resulting in an updated, more accurate estimate called "posterior belief". However, a crucial limitation of BP is that it only provides

---

* These authors contributed equally.
† Corresponding authors.

38th Conference on Neural Information Processing Systems (NeurIPS 2024).

point estimates for these posterior beliefs [7] and does not capture their potential uncertainty, leading to decisions unaware of the underlying data and model unreliability. For example, in a social network, a higher inferred probability of Alice's interest in sports over Bob's may lead to erroneous decisions if, in fact, Alice's inferred probability has high uncertainty and only a slightly higher interest.

To handle the uncertainty present in the beliefs, some well-known sampling-based uncertainty quantification (UQ) tools, such as Monte Carlo (MC) simulations, can provide unbiased UQ. However, these techniques are time-consuming for large-scale graphs and cannot ensure convergence within a reasonable time frame in practice [21]. Existing works [7, 35], based on Bayesian theory, have derived UQ methods with provable convergence and scalability by modeling beliefs as Dirichlet distributions and treating neighboring nodes as observations. However, this means that any neighbor of a node will necessarily reduce the uncertainty, even neighbors with noise or missing information. As illustrated in Figure 1, consider a user A with no preference information about being a music enthusiast, and A has two friends who are music enthusiasts (represented by a strong preference with Beta distribution parameters like $\mathcal{B}(3,1)$, where the parameters represent positive and negative preference counts). If A gains an additional friend D who has no preference information (represented by a uniform distribution $\mathcal{B}(1,1)$), the existing methods still reduce A's uncertainty, incorrectly suggesting increased confidence that A is a music enthusiast. This results in an underestimation of posterior uncertainty. Furthermore, none of existing works provided a theoretical relationship between the computed uncertainty and the expected model prediction error, making it difficult for decision-makers to understand and trust the UQ results.

To address the challenges mentioned above, we introduce Linear Uncertainty Propagation (*LinUProp*), a method to quantify the uncertainty in posterior beliefs resulting from multiple iterations of propagation of uncertainty in node priors, which offers the following advantages:

- Methodologically, we propose *LinUProp* (Sec. 3), a novel linear method that spreads the uncertainty from each node to the entire graph and additively aggregates this uncertainty to avoid underestimating posterior uncertainty. *LinUProp* offers interpretability due to the additive of neighbor uncertainty (Fig. 1), enabling tracking the contributions of other nodes to the computed uncertainty and allowing users to understand its sources.

- Theoretically, *LinUProp* possesses a closed-form solution, provable convergence related to the spectral radius of a matrix representing the dependencies between nodes (Sec. 4.1), and proven linear scalability (Appendix A.3). Moreover, by employing the bias-variance decomposition, we demonstrate that the posterior uncertainty is a **generalized variance component** of the expected model prediction error (Sec. 4.2).

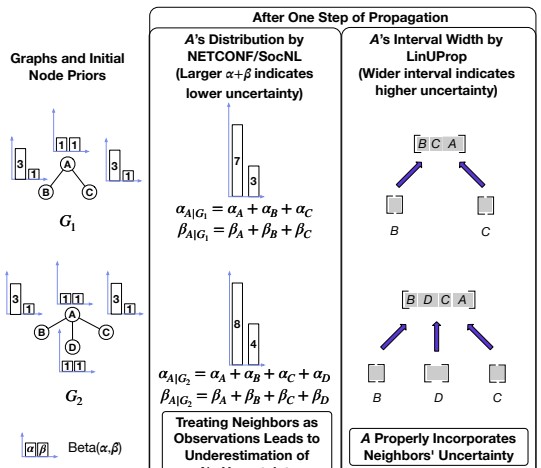

Figure 1: The impact of more neighbors on posterior uncertainty. NETCONF [7] or SocNL [35] underestimates posterior uncertainty of $A$ in $G_2$ by considering high-uncertainty node $D$ as neighbor that reduce uncertainty. *LinUProp* represents the uncertainty of a posterior using interval widths, where increased uncertainty from more neighbors leads to higher posterior uncertainty. The proposed interval ways can be interpret as a **generalized variance component** of the expected prediction error (Sec. 4.2). Furthermore, *LinUProp* is interpretable due to the additive of neighbor uncertainty (Eq. (12)).

- Experimentally, we demonstrate the following with *LinUProp*: (1) The uncertainty quantified by *LinUProp* is accurate due to its strong positive correlation with MC simulations (Fig. 7 in Appendix B.1); (2) *LinUProp* exhibits both fast convergence and linear scalability (Fig. 4); (3) *LinUProp* is interpretable (Fig. 3); (4) When applied to graph active learning guided by uncertainty, *LinUProp* achieves superior accuracy with a smaller budget compared to the baseline method that underestimates uncertainty (Figs. 5-6)

---

Project page including code at `https://github.com/chenghuaguo/LinUProp`

## 2 Preliminaries

**Belief Propagation (BP)** [27] iteratively passes messages on a graph to infer posterior distributions of variables on a graphical model. We focus on the uncertainty in the inferred posteriors used for node classification on Markov Random Field (MRF). Bayesian networks are another important type of graphical models that can be converted into MRF [27].

Consider a graph $\mathcal{G} = (\mathcal{V}, \mathcal{E})$ with $n$ random variable nodes, each with $k$ possible classes. The prior $\mathbf{e}_s$ for node $s$ is a $k$-dimension probability distribution and $e_s(i)$ specifies the prior probability for node $s$ belonging to class $i$. Each edge $(s, t)$ from $\mathcal{G}$ represents the dependencies between the two random variables $s$ and $t$. In particular, the dependency is represented by a pairwise potential function, which is a $k \times k$ *compatibility matrix* $\mathbf{H}$ and $H(i, j)$ denotes the degree of association between class $i$ of node $s$ and class $j$ of node $t$. We assume that $\mathbf{H}$ is a symmetric matrix, as in [11, 7]. Specifically, for binary classification, the compatibility matrix is assumed to be

$$\mathbf{H} = \begin{bmatrix} 0.5 + \epsilon & 0.5 - \epsilon \\ 0.5 - \epsilon & 0.5 + \epsilon \end{bmatrix}. \tag{1}$$

Similarly, an example form of $\mathbf{H}$ for multi-class problems is setting the diagonal elements to $\frac{1}{k} + (k - 1)\epsilon$ and the other elements to $\frac{1}{k} - \epsilon$. $|\epsilon|$ is typically close to 0, and a positive/negative $\epsilon$ specifies a homophily/heterophily relationship between any two connected nodes. Unlike existing methods that require all edges to have the same compatibility matrix, our proposed *LinUProp* can handle situations where each edge has a different compatibility matrix.

BP updates the $k$-dimensional *message* $\mathbf{m}_{ts}$ sent from node $t$ to node $s$ by:

$$m_{ts}(i) \leftarrow \sum_{j=1}^{k} H(i, j) e_t(j) \prod_{u \in \mathcal{N}(t) \setminus \{s\}} m_{ut}(j), i = 1, \ldots, k, \tag{2}$$

where $\mathcal{N}(t)$ is the set of neighboring nodes of $t$. Eq. (2) is applied iteratively until convergence or a maximum number of iterations is reached. Then the posterior $\mathbf{b}_s$ for node $s$ is

$$b_s(i) \leftarrow \frac{1}{Z_s} e_s(i) \prod_{t \in \mathcal{N}(s)} m_{ts}(i), \quad i = 1, \ldots, k, \tag{3}$$

where $Z_s$ is for normalization such that $\sum_{i=1}^{k} b_s(i) = 1$.

**Centered BP** [11] is a linearized version of Eqs. (2-3):

$$\hat{m}_{ts}(i) \leftarrow k \sum_{j=1}^{k} \hat{H}(i, j) \left( \hat{b}_t(j) - \frac{1}{k} \hat{m}_{st}(j) \right), \quad (4) \qquad \hat{b}_s(i) \leftarrow \hat{e}_s(i) + \frac{1}{k} \sum_{t \in \mathcal{N}(s)} \hat{m}_{ts}(i), \quad (5)$$

where $\hat{e}_s(i) = e_s(i) - \frac{1}{k}$, $\hat{b}_s(i) = b_s(i) - \frac{1}{k}$, $\hat{m}_{ts}(i) = m_{ts}(i) - 1$, $\hat{H}(i, j) = H(i, j) - \frac{1}{k}$, are the centralized version of the prior, belief, message, and compatibility matrix.

**NETCONF** [7] models both the belief and uncertainty of each node using a Dirichlet distribution. The certainty of a node is represented by the sum of its Dirichlet parameters, where a higher sum indicates greater certainty. Each node $u$ is initialized with a prior Dirichlet belief vector $\check{\mathbf{e}}_u$. The posterior Dirichlet belief $\check{\mathbf{b}}_u$ is updated using multinomial messages from its neighbors:

$$\check{\mathbf{b}}_u \leftarrow \check{\mathbf{e}}_u + \sum_{v \in \mathcal{N}(u)} \check{\mathbf{m}}_{vu}, \qquad \check{\mathbf{m}}_{vu} \leftarrow \mathbf{M} \left( \check{\mathbf{e}}_v + \sum_{w \in \mathcal{N}(v) \setminus u} \check{\mathbf{m}}_{wv} \right),$$

where $\check{\mathbf{m}}_{vu}$ is the message from node $v$ to $u$, and $\mathbf{M}$ is a modulation matrix derived from $\mathbf{H}$.

**Problem 1 (Quantifying Uncertainty in Posteriors).**

*Given*: (1) An undirected graph $\mathcal{G} = (\mathcal{V}, \mathcal{E})$ consisting of $n$ nodes and an adjacency matrix $\mathbf{A}$, (2) $|\mathcal{E}|$ compatibility matrices, where each matrix illustrates the dependency relationship between a pair of connected nodes, and (3) $n$ $k$-dimensional prior uncertainty vectors $\mathbb{e}_i (i = 1, \ldots, n)$, with each vector representing the uncertainty in prior beliefs for a node as interval widths across the $k$ classes. *Find* $n$ $k$-dimensional posterior uncertainty vectors $\mathbb{b}_i (i = 1, \ldots, n)$, where each vector represents the uncertainty of posterior beliefs for $k$ classes of a node based on interval width, with wider interval indicating higher uncertainty.

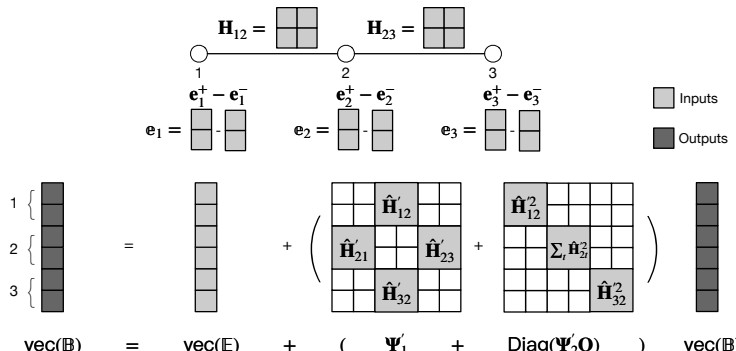

Figure 2: An illustration of *LinUProp* quantifying uncertainty in posterior beliefs for each node in a 3-node chain. *Inputs*: (1) Uncertainty in prior beliefs of each node represented as interval widths ($\mathbb{e}_1$, $\mathbb{e}_2$, $\mathbb{e}_3$) (2) Edge potentials ($\mathbf{H}_{12}$ and $\mathbf{H}_{23}$, with $\mathbf{H}_{21} = \mathbf{H}_{12}$ and $\mathbf{H}_{32} = \mathbf{H}_{23}$ due to the undirected graph). *Outputs*: uncertainty in posterior beliefs of each node also represented as interval widths. *LinUProp* can set a different compatibility matrix for each edge, allowing it to handle edge-dependent potentials, while previous methods cannot do this.

## 3   Linear Bound Propagation

Recall that our objective is to quickly quantify the uncertainty in posterior beliefs and not assume that neighbors necessarily reduce uncertainty, in order to avoid uncertainty underestimation. Interval arithmetic [6] is a non-probabilistic method for quantifying uncertainty without assumptions regarding neighbors. However, it is unclear how to directly apply interval arithmetic rules to Eqs. (2-3) of BP while meeting the above objectives.

A solution to these challenges lies in linearization. The linearized BP is more amenable to interval arithmetic due to its exclusive reliance on interval addition, sidestepping the interval multiplication that may underestimate uncertainty. Consequently, this facilitates the derivation of closed-form solution, endowing the method with linear scalability, interpretability, and guaranteed convergence.

We will next introduce *LinUProp* and its iterative variant, which is computationally efficient in practical applications. Prior to that, we need some additional notation pertinent to *LinUProp*. Let vec denotes the operation that vertically concatenates the rows of a given matrix into a single column vector, and let Diag denote the transformation of an $nk \times k$ block matrix into a block diagonal matrix. $\mathbb{E} = [\mathbb{e}_1, \mathbb{e}_2, \cdots, \mathbb{e}_n]^T$ and $\mathbb{B} = [\mathbb{b}_1, \mathbb{b}_2, \cdots, \mathbb{b}_n]^T$ respectively represent the uncertainties of prior and posterior beliefs for all $n$ nodes, each of dimension $n \times k$. $\mathbf{Q}$ is an $nk \times k$ matrix formed by vertically stacking $n$ identity matrices, each of size $k \times k$. $\mathbf{\Psi}_1^{'}$ and $\mathbf{\Psi}_2^{'}$ are $nk \times nk$ block matrices formed from $\hat{\mathbf{H}}_{st}^{'}$ and $\hat{\mathbf{H}}_{st}^{'2}$, respectively. For any given edge $(s, t)$, $\hat{\mathbf{H}}_{st}^{'}$ denotes the centralized compatibility matrix corresponding to that edge, with entries $\hat{H}_{st}^{'}$ defined as $\hat{H}_{st}^{'}(i, j) = |H_{st}(i, j) - \frac{1}{k}|$. In cases where the edge $(s, t)$ does not exist, the matrix defaults to zero.

$$\text{Diag}\left(\begin{bmatrix} \mathbf{L_1} \\ \vdots \\ \mathbf{L_n} \end{bmatrix}\right) = \begin{bmatrix} \mathbf{L_1} & & \\ & \ddots & \\ & & \mathbf{L_n} \end{bmatrix} \quad \mathbf{\Psi}_1^{'} = \begin{bmatrix} \hat{\mathbf{H}}_{11}^{'} & \cdots & \hat{\mathbf{H}}_{1n}^{'} \\ \vdots & \ddots & \vdots \\ \hat{\mathbf{H}}_{n1}^{'} & \cdots & \hat{\mathbf{H}}_{nn}^{'} \end{bmatrix} \quad \mathbf{\Psi}_2^{'} = \begin{bmatrix} \hat{\mathbf{H}}_{11}^{'2} & \cdots & \hat{\mathbf{H}}_{1n}^{'2} \\ \vdots & \ddots & \vdots \\ \hat{\mathbf{H}}_{n1}^{'2} & \cdots & \hat{\mathbf{H}}_{nn}^{'2} \end{bmatrix}$$

**Theorem 3.1 (*LinUProp*).** *For a multi-class node classification task on an MRF, given the matrix $\mathbb{E}$ which represents the prior belief uncertainty of all nodes, matrices $\mathbf{\Psi}_1^{'}$ and $\mathbf{\Psi}_2^{'}$ which denote the dependencies among these nodes. The posterior belief uncertainty for all nodes, represented by $\mathbb{B}$ and in terms of interval widths, is determined by the linear equation system:*

$$\text{vec}(\mathbb{B}) = \text{vec}(\mathbb{E}) + \left(\mathbf{\Psi}_1^{'} + \text{Diag}\left(\mathbf{\Psi}_2^{'}\mathbf{Q}\right)\right)\text{vec}(\mathbb{B}). \tag{6}$$

Figure (2) illustrates Eq. (6) using a 3-node chain as an example. From this, it can be observed that *LinUProp* can be implemented using matrix multiplication without computing the uncertainty propagated between each pair of nodes. In practice, a more effective strategy for computing the

posterior belief interval widths $\text{vec}(\mathbb{B})$ is to use an iterative update version of *LinUProp*:

$$\text{vec}(\mathbb{B})^{(l+1)} = \text{vec}(\mathbb{E}) + \left( \mathbf{\Psi}_1^{'} + \text{Diag}\left( \mathbf{\Psi}_2^{'} \mathbf{Q} \right) \right) \text{vec}(\mathbb{B})^{(l)}, \tag{7}$$

where $l$ denotes the iteration round and $\text{vec}(\mathbb{B})^{(0)}$ can be set to $\text{vec}(\mathbb{E})$. Although Eq. (6) is similar in form to another method, *LinBP* (Eq. (13), [11]), which is a point estimation inference approach rather than a UQ method, simply replacing prior and posterior beliefs in *LinBP* with interval widths does not yield *LinUProp*. *LinUProp* is specifically designed to quantify uncertainty through derivations involving the upper and lower bounds of messages and beliefs. For detailed derivations, please refer to Appendix A.1. Its guaranteed convergence is proven in Sec. 4.1. The proof of its linear scalability is provided in Appendix A.3, showing that the time complexity per iteration is $\mathcal{O}(|\mathcal{V}|)$ when $|\mathcal{V}| > |\mathcal{E}|$, otherwise $\mathcal{O}(|\mathcal{E}|)$.

## 4 Analysis

We present theoretical analyses of *LinUProp*, including the derivation of a closed-form solution, the proof of its convergence, and the provision of an interpretable approach to the uncertainty quantified by *LinUProp*. Furthermore, by employing the bias-variance decomposition, we addressed the question of what the uncertainty computed by *LinUProp* represents and established the connection between this uncertainty and the expected prediction error.

### 4.1 Theoretical Analysis of *LinUProp*

**Closed-form solution.** By simplifying Eq. (6), we can derive a closed-form solution for *LinUProp*:

$$\text{vec}(\mathbb{B}) = \underbrace{(\mathbf{I} - \mathbf{\Psi}_1^{'} - \text{Diag}(\mathbf{\Psi}_2^{'} \mathbf{Q}))^{-1}}_{\mathbf{F}'} \text{vec}(\mathbb{E}), \tag{8}$$

where $\mathbf{I}$ represents the identity matrix. This closed-form solution is primarily used for subsequent theoretical analysis; in practice, the iterative version Eq. (7) is employed, which does not require the computation of the matrix inverse.

**Convergence.** Let $\mathbf{\Psi}_1^{'} + \text{Diag}(\mathbf{\Psi}_2^{'} \mathbf{Q})$ be denoted as $\mathbf{T}$. The sufficient and necessary criteria for the convergence of *LinUProp* is that the spectral radius of $\mathbf{T}$ is less than 1:

$$\textit{LinUProp} \text{ converges} \iff \rho(\mathbf{T}) < 1 \tag{9}$$

*Proof.* The closed-form solution of *LinUProp* (Eq. (8)) conforms to a general linear equation system, $\mathbf{y} = (\mathbf{I} - \mathbf{P})^{-1} \mathbf{x}$, where $\mathbf{y}$, $\mathbf{P}$, and $\mathbf{x}$ are generic terms. Such linear equation systems can be solved by the Jacobi method [29], which converges if and only if the spectral radius of $\mathbf{P}$ is less than 1. $\square$

From the above convergence condition, *LinUProp* has a limitation: in large graphs with strong global (most edges) homophily/heterophily, $\rho(\mathbf{T})$ may be large, leading to non-convergence. However, *LinUProp* can still converge if such strong homophily/heterophily is only local (a few edges).

**Interpretability.** When *LinUProp* converges, with $\rho(\mathbf{T}) < 1$, we can expand the closed-form solution Eq. (8) using the Neumann series, yielding:

$$\text{vec}(\mathbb{B}) = \left( \mathbf{I} + \mathbf{T} + \mathbf{T}^2 + \cdots \right) \text{vec}(\mathbb{E}). \tag{10}$$

The uncertainty of the posterior belief for a certain class of a node, $\text{vec}(\mathbb{B})_v$ can be expanded as

$$\text{vec}(\mathbb{B})_v = \text{vec}(\mathbb{E})_v + \mathbf{T}_v \text{vec}(\mathbb{E}) + \left( \mathbf{T}^2 \right)_v \text{vec}(\mathbb{E}) + \cdots, \tag{11}$$

where $\mathbf{T}_v \text{vec}(\mathbb{E})$, $\left( \mathbf{T}^2 \right)_v \text{vec}(\mathbb{E})$, etc. can be expanded to $\sum_w \mathbf{T}_{v,w} \text{vec}(\mathbb{E})_w$, $\sum_w \left( \mathbf{T}^2 \right)_{v,w} \text{vec}(\mathbb{E})_w$ and so on, the subscript $v$ represents the $v$-th row of matrices $\mathbf{T}, \mathbf{T}^2, \cdots$, while the subscript $w$ represents the $w$-th column of matrices $\mathbf{T}, \mathbf{T}^2, \cdots$ and the $w$-th component of $\text{vec}(\mathbb{E})$. Finally, we derive the contribution of the prior uncertainty of a certain class of any other variable node $\text{vec}(\mathbb{E})_w$ towards the posterior belief uncertainty $\text{vec}(\mathbb{B})_v$ as

$$c_{w \to v} = \mathbf{T}_{v,w} \text{vec}(\mathbb{E})_w + \left( \mathbf{T}^2 \right)_{v,w} \text{vec}(\mathbb{E})_w + \left( \mathbf{T}^3 \right)_{v,w} \text{vec}(\mathbb{E})_w + \cdots \tag{12}$$

This linear equation facilitates simple explanations of the uncertainty computed by *LinUProp*. Since $\rho(\mathbf{T}) < 1$ ensures convergence, the influence of higher-order terms, such as $\mathbf{T}^3$ and $\mathbf{T}^4$, decays rapidly. Therefore, in practice, considering only the first few terms often provides an accurate and interpretable estimate of uncertainty contributions.

## 4.2 Theoretical Connection with Bias-Variance Decomposition

To delve into the theoretical understanding of the uncertainty in the node's posterior belief, we employed the bias-variance decomposition [2], an effective tool for analyzing prediction errors and decomposing model uncertainty [41]. We demonstrated that the posterior uncertainty computed by *LinUProp* is a *generalized variance component* of the expected model prediction error, giving *LinUProp* a solid theoretical basis. This relationship between the uncertainty computed by UQ methods and the expected model prediction error has seldom been explored by existing work [7, 36, 35]. To begin with, we first derive a linearized BP suitable for edges with distinct potential functions to facilitate bias-variance decomposition.

From the update equation of message $\hat{\mathbf{m}}_{ts}$ as shown in Eq. (4), we can derive the update rule for the reverse message $\hat{\mathbf{m}}_{st}$ and substitute it back. This leads us to the stable message $\hat{\mathbf{m}}_{ts}$ when the algorithm converges. By inserting the stable message into Eq. (5) and simplifying, we can derive the closed-form solution for posterior beliefs where each edge has a distinct potential function (detailed proof in Appendix A):

$$\text{vec}(\hat{\mathbf{B}}) = \underbrace{(\mathbf{I} - \boldsymbol{\Psi}_1 + \text{Diag}(\boldsymbol{\Psi}_2\mathbf{Q}))^{-1}}_{\mathbf{F}} \text{vec}(\hat{\mathbf{E}}), \tag{13}$$

$$\boldsymbol{\Psi}_1 = \begin{bmatrix} \hat{\mathbf{H}}_{11} & \cdots & \hat{\mathbf{H}}_{1n} \\ \vdots & \ddots & \vdots \\ \hat{\mathbf{H}}_{n1} & \cdots & \hat{\mathbf{H}}_{nn} \end{bmatrix} \quad \boldsymbol{\Psi}_2 = \begin{bmatrix} \hat{\mathbf{H}}_{11}^2 & \cdots & \hat{\mathbf{H}}_{1n}^2 \\ \vdots & \ddots & \vdots \\ \hat{\mathbf{H}}_{n1}^2 & \cdots & \hat{\mathbf{H}}_{nn}^2 \end{bmatrix}$$

where vec and Diag are the same as those in Eq. (6). $\hat{\mathbf{B}} = [\hat{\mathbf{b}}_1, \hat{\mathbf{b}}_2, \cdots, \hat{\mathbf{b}}_n]^T$ and $\hat{\mathbf{E}} = [\hat{\mathbf{e}}_1, \hat{\mathbf{e}}_2, \cdots, \hat{\mathbf{e}}_n]^T$ are $n \times k$ matrices composed of the centralized posterior and prior beliefs of all nodes. $\mathbf{Q}$ is an $nk \times k$ matrix formed by vertically stacking $n$ identity matrices, each of size $k \times k$. $\boldsymbol{\Psi}_1$ and $\boldsymbol{\Psi}_2$ are $nk \times nk$ block matrices formed from $\hat{\mathbf{H}}_{st}$ and $\hat{\mathbf{H}}_{st}^2$, respectively. For any given edge $(s, t)$, $\hat{\mathbf{H}}_{st}$ denotes the centralized compatibility matrix corresponding to that edge, with entries $\hat{\mathbf{H}}_{st}$ defined as $\hat{H}_{st}(i, j) = H_{st}(i, j) - \frac{1}{k}$. If the edge $(s, t)$ does not exist, the matrix defaults to zero.

This closed-form solution shows that the posterior belief can be regarded as a linear model of the prior belief. Specifically, consider the element $\text{vec}(\hat{\mathbf{B}})_v$. This element represents the centralized posterior belief of a specific class corresponding to a particular node. It is formulated as a linear combination of all centralized prior beliefs, represented by $\mathbf{F}_v \text{vec}(\hat{\mathbf{E}})$. Here, $\mathbf{F}_v$ denotes the $v$-th row of $\mathbf{F}$. This linear equation enables us to derive a bias-variance decomposition of the uncertainty in the posterior belief. We define $h(\hat{\mathbf{E}})$ as the centralized posterior belief corresponding to a specific class for a particular node in an unknown true linear model, and represent the centralized belief about $\hat{\mathbf{E}}$ computed using Eq. (13) as $\text{vec}(\hat{\mathbf{B}})_v$. Then, the expected model prediction error can be decomposed into bias and variance as

$$\mathbb{E}\left[\left(h(\hat{\mathbf{E}}) - \text{vec}(\hat{\mathbf{B}})_v\right)^2\right] = \underbrace{\left(h(\hat{\mathbf{E}}) - \mathbb{E}\left[\text{vec}(\hat{\mathbf{B}})_v\right]\right)^2}_{(\text{Bias})^2} + \underbrace{\mathbb{E}\left[\left(\text{vec}(\hat{\mathbf{B}})_v - \mathbb{E}\left[\text{vec}(\hat{\mathbf{B}})_v\right]\right)^2\right]}_{\text{Variance}}. \tag{14}$$

Expanding the variance term yields

$$\mathbb{E}\left[\left(\text{vec}(\hat{\mathbf{B}})_v - \mathbb{E}[\text{vec}(\hat{\mathbf{B}})_v]\right)^2\right] = \mathbf{F}_v \Sigma_{\text{vec}(\hat{\mathbf{E}})} \mathbf{F}_v^T, \tag{15}$$

where $\Sigma_{\text{vec}(\hat{\mathbf{E}})}$ is the covariance matrix of $\text{vec}(\hat{\mathbf{E}})$ (proof in Appendix A.4). If we assume that the priors of different nodes and class probabilities within each node's prior are nearly independent, $\Sigma_{\text{vec}(\hat{\mathbf{E}})}$ can be approximated as a diagonal matrix. Consequently, the sign of the elements in $\mathbf{F}_v$

will not affect the result of Eq. (15). Ignoring the small term $\text{Diag}(\cdot)$, the variance component can be approximated as $\mathbf{F}'_v \Sigma_{\text{vec}(\hat{\mathbf{E}})} \mathbf{F}'^T_v$. The posterior interval width for $v$ computed by *LinUProp* is $\text{vec}(\mathbb{B})_v = \mathbf{F}'_v \text{vec}(\mathbb{E})$, leading to $\text{vec}(\mathbb{B})_v (\text{vec}(\mathbb{B})^T)_v = \mathbf{F}'_v (\text{vec}(\mathbb{E}) \text{vec}(\mathbb{E})^T) \mathbf{F}'^T_v$. This implies that the variance component is a **special case** of *LinUProp* where the outer product of the prior interval width $\text{vec}(\mathbb{E}) \text{vec}(\mathbb{E})^T$ is the covariance matrix $\Sigma_{\text{vec}(\hat{\mathbf{E}})}$. **In other words**, the uncertainty computed by *LinUProp* is a *generalized variance component* of the expected prediction error.

## 5 Experiments

We begin with a simple case study to illustrate the correctness and interpretability of quantified uncertainty. We then present quantitative evidence of our method's ability to accurately quantify uncertainty, and demonstrate the convergence and scalability of our methods on real-world datasets. Finally, we compare the efficacy of *LinUProp* to other competitors in graph active learning tasks.

### 5.1 Case Study on a Simple Graph

**Correctness of quantified uncertainty.** To verify the correctness of the uncertainty quantified by *LinUProp* for posterior beliefs, we first qualitatively analyze the belief bounds on a simple graph. Specifically, we conduct the experiment on a 4×4 grid, as shown in Figure 3(a), with priors set to

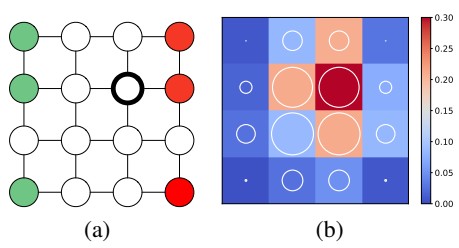

(a)                          (b)

Figure 3: Case Study. (a) A 4×4 grid with two classes. The nodes colored in red and green are labeled, while the rest are unlabeled. The bold unlabeled node indicates the node we aim to explain the source of uncertainty. (b) An interpretation of the uncertainty in the belief of the bold node computed by *LinUProp*. The **colors** represent the contribution of each node to the uncertainty of the bold node, with warmer colors indicating more significant contributions; the **radius** of the white circles indicates the belief bound width computed by *LinUProp* for each node, with a larger radius indicating higher uncertainty in the beliefs.

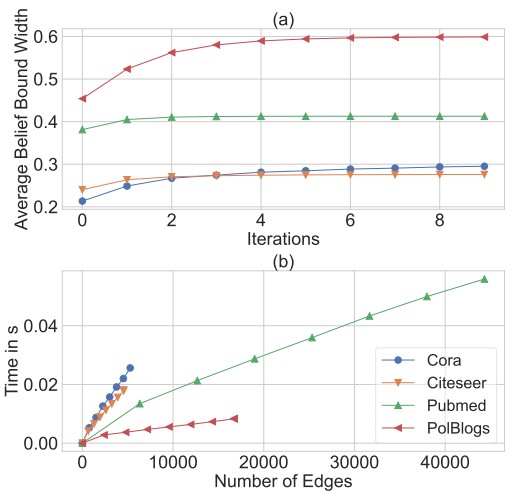

Figure 4: (a) Convergence of average belief bound width by *LinUProp*. (b) Scalability. Each data point represents the running time of *LinUProp* for 10 iterations with a certain number of edges. The y-axis is the running time in seconds.

$\mathcal{B}(9, 1)$ (red nodes) or $\mathcal{B}(1, 9)$ (green node) for labeled nodes, and $\mathcal{B}(1, 1)$ for the unlabeled nodes. We use Beta distributions as priors in binary classification because they are a natural choice for binary random variables [2]. For each variable node, the prior interval width for each class is set to twice the standard deviation of the corresponding prior Beta distribution. For each edge, the compatibility matrix is set as in Eq. (1), with $\epsilon = 0.1$.

Figure 3(b) shows that the more labeled nodes around each node $s$, the smaller the uncertainty in the belief; the more unlabeled nodes around $s$, the larger the uncertainty in the belief, which is intuitive. In addition to the qualitative analysis mentioned above, we also compared the uncertainty quantified by *LinUProp* with that quantified by MC simulations. MC simulations are adopted as the ground-truth due to its ability to provide accurate approximations through a sufficient amount of sampling [30], which are feasible for small-scale graphs. The experimental results show a strong positive correlation between the uncertainties quantified by the two methods (PCC=0.9084), which can be found in Appendix B.1.

**Interpretability.** By Eq. (12), we can compute the contribution of any node $t$ to the uncertainty in the belief of node $s$ computed by *LinUProp*. Figure 3(b) shows that the main source of uncertainty for the bold node is itself because it is an unlabeled node. The unlabeled nodes in the neighborhood are the secondary sources, and the unlabeled nodes within 2 hops are the tertiary sources. It demonstrates that the uncertainty in the beliefs quantified by *LinUProp* has good interpretability, which can enhance users' trust in UQ results computed by *LinUProp*.

## 5.2 Experiments on Real Data

We validate the properties of *LinUProp* on three popular citation networks (Cora, Citeseer, and PubMed) [18] and a political blog hyperlink network (PolBlogs) [1]. For further details regarding these datasets and experimental configurations, please refer to Appendix B.2. Our experimental findings can be summarized into the following three aspects.

**Convergence.** We set node priors based on classification type: Beta distributions for binary and Dirichlet distributions for multi-class ($k$ classes), both using parameter vector $\boldsymbol{\alpha}$ of length $k$. In datasets, 30% of nodes are randomly labeled; if labeled as class $i$, $\alpha_i = 10$, otherwise, entries are 1. Unlabeled nodes have $\boldsymbol{\alpha} = \mathbf{1}$. Prior interval widths for each class are twice the standard deviation of the distribution, capturing uncertainty by representing the interval as mean±std.

To simulate the diversity of dependencies between nodes, each edge's $\epsilon$ is randomly selected from {1e-4, 5e-4, 1e-3, 5e-3, 1e-2} and linked to a $k \times k$ compatibility matrix. Diagonal elements are $\frac{1}{k} + (k-1)\epsilon$, and other elements are $\frac{1}{k} - \epsilon$. We monitor *LinUProp*'s convergence (iterative version in Eq. (7)) by measuring the average belief bound width, $\sum_{p=1}^{n} \sum_{q=1}^{k} \mathbb{B}(p,q)/(n*k)$.

Figure 4(a) shows that *LinUProp* converges within 10 iterations across all four datasets, demonstrating its rapid convergence. Using an iterative update version of *LinUProp*, the posterior belief bound width for each node increases over iterations due to uncertainty propagation from other nodes.

**Scalability.** We initialize the prior interval width for nodes and the compatibility matrix for edges following the same procedure as in the convergence experiment. Then we uniformly sampled different numbers of edges from four datasets and recorded the running time of *LinUProp* for 10 iterations. Figure 4(b) shows that the running time scales linearly in the number of edges. For more details on the runtime comparison between NETCONF and *LinUProp*, please refer to Appendix B.4.

**Effectiveness.** For large-scale graphs, MC sampling is computationally impractical due to the significantly increased time required for each sample. Therefore, we evaluate *LinUProp*'s effectiveness using active learning as a downstream task. Specifically, we use uncertainty-based sampling to select the next node for label acquisition, as detailed in [23]. In this context, users seek a labeled dataset with minimal uncertainty, maximum accuracy, and minimal labeling budget.

We simulate a realistic scenario as in [40], using training set $\mathcal{V}_{train}$, validation set $\mathcal{V}_{val}$, test set $\mathcal{V}_{test}$, and unlabeled pool $\mathcal{V}_{ulp}$ (node numbers in Table 1, Appendix B.2), with query batch size $b$. Initially, training set nodes are labeled and others are unlabeled. Node priors follow the convergence experiment procedure. Edge compatibility matrices match the correctness experiment, enabling comparisons with existing methods that cannot handle diverse potential functions. Due to noisy labeling, the unlabeled pool remains unchanged to allow re-labeling [31].

For each selected node $s$, we update its Dirichlet prior by incrementing the parameter for the given label class by 1. After each labeling iteration, all nodes undergo inference. Noisy labels may not guarantee improved inference accuracy, so we use the iteration yielding the highest validation accuracy to evaluate the test set, which determines the test accuracy for the current labeling budget. MC-based methods are unsuitable for active learning as it requires time-consuming sampling for each iteration. We use the following strategies in each iteration:

- **Random:** Select $b$ nodes randomly.

- **Least Confidence (LC):** Calculate uncertainty as $U_{LC}(s) = 1 - \mathrm{argmax}_i\, b_s(i)$.

- **Entropy:** Calculate uncertainty as $U_{Entropy}(s) = -\sum_i b_s(i) \log b_s(i)$.

- **Certainty Score (CS):** Let $\boldsymbol{\breve{b}_s}$ be the parameter vector of the posterior Dirichlet distribution of node $s$ inferred by NETCONF [7]. Uncertainty is $U_{CS}(s) = -\sum_i \breve{b}_s(i)$. Due to assumptions on distribution forms of priors, messages, and posterior beliefs, only applicable with NETCONF.

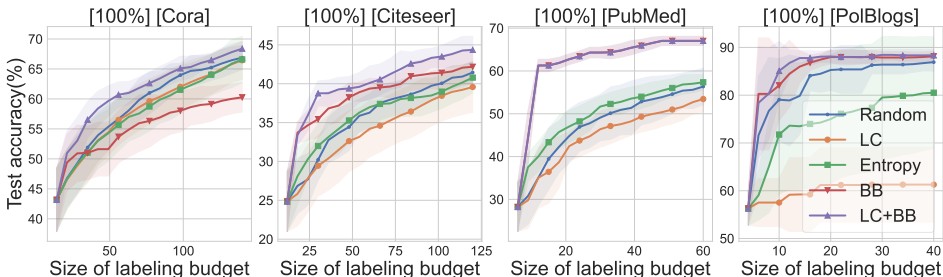

Figure 5: Test accuracy for varying labeling budgets with **BP** inferring posterior beliefs. Each subplot title includes two components, which represent the labeling accuracy and dataset. Each column corresponds to a dataset. In each subplot, the node selection strategies based on *LinUProp* and its variant are represented by red ▼ and purple ▲. Under the same labeling budget, the higher the test accuracy, the better.

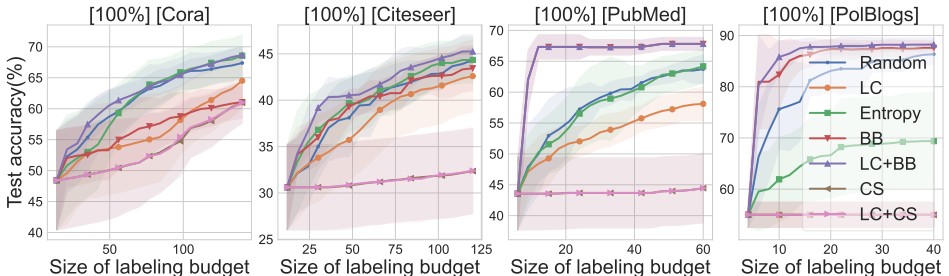

Figure 6: Test accuracy for varying labeling budgets with **NETCONF** inferring posterior beliefs. Each subplot title includes two components, which represent the labeling accuracy and dataset. Each column corresponds to a dataset. In each subplot, the node selection strategies based on *LinUProp* and its variants are represented by ▼ and purple ▲, while the strategies based on the native UQ method in NETCONF and its variants are represented by brown ◄ and pink ►. NETCONF-based strategies (CS/LC+CS) often prioritize labeling low-degree nodes due to their assumption that neighbors serve as evidence, leading to persistent high uncertainty in most nodes and slowly increasing accuracy. Under the same labeling budget, the higher the test accuracy, the better.

- **Belief Bound (BB):** Based on *LinUProp* with the same prior bound width setting as in the convergence experiment, uncertainty is $U_{BB}(s) = \mathbb{b}_s$.
- **LC+CS:** Perform Min-Max normalization on $U_{LC}(s)$ and $U_{CS}(s)$ to obtain $U_{LC}^{norm}(s)$ and $U_{CS}^{norm}(s)$. Uncertainty is $U_{LC+CS}(s) = U_{LC}^{norm}(s) + U_{CS}^{norm}(s)$.
- **LC+BB:** Combine LC and BB. Compute uncertainty as: $U_{LC+BB}(s) = U_{LC}^{norm}(s) + U_{BB}^{norm}(s)$.

We set query batch size $b = 2k$ ($k$ is the number of classes) and maximum labeling budget to $20b$. We evaluate node selection strategy performance with annotator labeling accuracy at 70%, 80%, 90%, and 100%. To reduce randomness in results, we repeat each method ten times (re-partitioning datasets and changing random seeds) and record the mean of test set inference accuracies. To demonstrate labeling budget impact, we conduct experiments evaluating various node selection strategies under different labeling budgets on four datasets, varying the budget from $2b$ to $20b$.

As shown in Figures 5 and 6, whether using BP or NETCONF for inference, the test accuracy of the *LinUProp*-based node selection strategy (BB) and its variations (LC+BB) generally grow faster than other baselines as the budget increases. This is especially the case when compared to the native UQ method in NETCONF (CS) and its variations (LC+CS). As shown in Figure 6, strategies based on NETCONF (CS/LC+CS) prioritize labeling low-degree nodes due to their inherent assumption that neighbors necessarily reduce the uncertainty. As a result, nodes with many neighbors are often mistakenly viewed as having low uncertainty and are left unlabeled, which further leads to high uncertainty in the majority of nodes. This phenomenon is more pronounced in graphs with a large number of low-degree nodes, like PolBlogs, leading to a very slow increase in accuracy. Experiments with a lower labeling accuracy yielded similar conclusions, as shown in Appendix B.3.1.

We also evaluate all strategies under a fixed labeling budget of $20b$ for a fair comparison, with results displayed in Appendix B.3.2. The conclusion is that whether using BP or NETCONF for inference,

BB and LC+BB outperform baselines across different labeling accuracies and datasets in most cases. From the results in active learning tasks, we see node selection strategies guided by *LinUProp*'s UQ results achieves higher labeling accuracy with lower labeling budget. This indicates that uncertainty quantified by *LinUProp* is accurate, effective, and insensitive to the inference method.

## 6 Related Work

**Uncertainty in posterior belief.** In addition to traditional UQ methods like MC simulations, which require extensive sampling, some research focuses on modeling uncertainty in posterior belief by deriving closed-form solutions to incorporate uncertainty into inference results. Existing methods [7, 36, 35] exhibit scalability but limited as they assume uniform potential functions across all edges. Furthermore, these methods with closed-form solutions assume that any neighbor of a node will necessarily reduce the uncertainty even neighbors affected by noise or lacking information, potentially leading to uncertainty underestimation. Moreover, few existing studies have theoretically linked calculated uncertainty to the expected model prediction error, making it difficult for decision-makers to understand and trust the UQ results.

Existing works [22, 20] utilize bound propagation without the aforementioned assumptions regarding neighbors. However, they focus on quantifying the error between the posterior beliefs and the true marginal probabilities of the variable nodes, rather than quantifying uncertainty. This fundamentally differs from *LinUProp*, which quantifies the posterior uncertainty as the generalized variance component of the expected prediction error.

**Human understanding of PGM inference.** Humans are unlikely to adopt inference outcomes without reasonable interpretation [33]. In [24, 32], the authors studied explainable Bayesian networks. Recently, explanations of inference on Bayesian network and MRF were found by differentiation [3, 5], so that a set of important network parameters (potentials) can explain the changes in the inferred posterior distribution of a target variable. Interpretable graphical models are also studied under the hood of topic models [4] or GNN [38]. None of the aforementioned studies can provide an explanation for the *uncertainty* in the inference results.

## 7 Conclusion

In this paper, we proposed *LinUProp*, a UQ method for graphical model inference that utilizes a novel linear propagation of uncertainty to model uncertainty among related nodes additively. *LinUProp* provides linear scalability, guaranteed convergence, and is interpretable. Unlike its competitors, *LinUProp* does not assume neighbors necessarily reduce uncertainty and thus avoids uncertainty underestimation. To gain deeper insights, we decompose the expected prediction error of the graphical model and prove that the uncertainty computed by *LinUProp* is the generalized variance component of the decomposition. Experimental analysis shows *LinUProp* possesses aforementioned properties and outperforms competitors in downstream tasks. However, the study has not yet explored the interpretability of uncertainty with human involvement in real-world decision-making processes, an area we aim to address in future research. We would also like to apply *LinUProp* to more applications of graphical models to demonstrate its utility in the future.

## Acknowledgments and Disclosure of Funding

Chenghua Guo and Xi Zhang were supported by the Natural Science Foundation of China (No. 62372057). This material is based upon work supported by the National Science Foundation under Grant Number 2008155. Sihong Xie was supported in part by the National Key R&D Program of China (Grant No. 2023YFF0725001), the Guangzhou-HKUST(GZ) Joint Funding Program (Grant No. 2023A03J0008), and Education Bureau of Guangzhou Municipality. Qi Li was supported in part by the National Science Foundation under NSF Grants IIS 2007941. Chao Chen was supported by the National Key Research and Development Program of China (No. 2023YFB3106504), and Pengcheng-China Mobile Jointly Funded Project (No. 2024ZY2B0050).

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

# A Proofs

## A.1 Proof of Theorem 3.1

Let vec denotes the operation that vertically concatenates the rows of a given matrix into a single column vector, and let Diag denote the transformation of an $nk \times k$ block matrix into a block diagonal matrix. $\mathbb{E} = [\mathbb{e}_1, \mathbb{e}_2, \cdots, \mathbb{e}_n]^T$ and $\mathbb{B} = [\mathbb{b}_1, \mathbb{b}_2, \cdots, \mathbb{b}_n]^T$ respectively represent the uncertainties of prior and posterior beliefs for all $n$ nodes, each of dimension $n \times k$. $\mathbf{Q}$ is an $nk \times k$ matrix formed by vertically stacking $n$ identity matrices, each of size $k \times k$. $\Psi_1$ and $\Psi_2$ are $nk \times nk$ block matrices formed from $\hat{\mathbf{H}}'_{st}$ and $\hat{\mathbf{H}}'^2_{st}$, respectively. For any given edge $(s,t)$, $\hat{\mathbf{H}}'_{st}$ denotes the centralized compatibility matrix corresponding to that edge, with entries $\hat{\mathbf{H}}'_{st}$ defined as $\hat{H}'_{st}(i,j) = |H_{st}(i,j) - \frac{1}{k}|$. In cases where the edge $(s,t)$ does not exist, the matrix defaults to zero.

$$\text{Diag}\left(\begin{bmatrix}\mathbf{L_1} \\ \vdots \\ \mathbf{L_n}\end{bmatrix}\right) = \begin{bmatrix}\mathbf{L}_1 & & \\ & \ddots & \\ & & \mathbf{L}_n\end{bmatrix} \quad \Psi'_1 = \begin{bmatrix}\hat{\mathbf{H}}'_{11} & \cdots & \hat{\mathbf{H}}'_{1n} \\ \vdots & \ddots & \vdots \\ \hat{\mathbf{H}}'_{n1} & \cdots & \hat{\mathbf{H}}'_{nn}\end{bmatrix} \quad \Psi'_2 = \begin{bmatrix}\hat{\mathbf{H}}'^2_{11} & \cdots & \hat{\mathbf{H}}'^2_{1n} \\ \vdots & \ddots & \vdots \\ \hat{\mathbf{H}}'^2_{n1} & \cdots & \hat{\mathbf{H}}'^2_{nn}\end{bmatrix}$$

**Theorem 3.1** (*LinUProp*). *For a multi-class node classification task on an MRF, given the matrix $\mathbb{E}$ which represents the prior belief uncertainty of all nodes, matrices $\Psi'_1$ and $\Psi'_2$ which denote the dependencies among these nodes. The posterior belief uncertainty for all nodes, represented by $\mathbb{B}$ and in terms of interval widths, is determined by the linear equation system:*

$$\text{vec}(\mathbb{B}) = \text{vec}(\mathbb{E}) + \left(\Psi'_1 + \text{Diag}\left(\Psi'_2\mathbf{Q}\right)\right)\text{vec}(\mathbb{B}). \tag{6}$$

*Proof.* Based on the linear approximation of BP messages defined in Eq. (4), we can approximate the lower and upper bounds of BP messages using interval arithmetic rules [6]:

$$\hat{m}^-_{ts}(i) \leftarrow k\sum_j \hat{H}_{st}(i,j)\hat{b}^{(1)}_{t\backslash s}(j), \tag{16}$$

$$\hat{m}^+_{ts}(i) \leftarrow k\sum_j \hat{H}_{st}(i,j)\hat{b}^{(2)}_{t\backslash s}(j), \tag{17}$$

$$\hat{b}^{(1)}_{t\backslash s}(j) = \begin{cases} \hat{b}^-_t(j) - \frac{1}{k}\hat{m}^+_{st}(j), & \hat{H}(i,j) > 0, \\ \hat{b}^+_t(j) - \frac{1}{k}\hat{m}^-_{st}(j), & \hat{H}(i,j) < 0, \end{cases} \tag{18}$$

$$\hat{b}^{(2)}_{t\backslash s}(j) = \begin{cases} \hat{b}^-_t(j) - \frac{1}{k}\hat{m}^+_{st}(j), & \hat{H}(i,j) < 0, \\ \hat{b}^+_t(j) - \frac{1}{k}\hat{m}^-_{st}(j), & \hat{H}(i,j) > 0. \end{cases} \tag{19}$$

The idea behind Eqs. (16-19) is that the determination of the lower and upper bounds of $\hat{m}_{ts}(i)$ is contingent upon the sign of $\hat{H}_{st}(i,j)$. Specifically, when determining the lower bound of $\hat{m}_{ts}(i)$, if $\hat{H}_{st}(i,j) < 0$, then $\hat{b}_{t\backslash s}(j)$ should be maximized, i.e., taking its upper bound $\hat{b}^+_t(j) - \frac{1}{k}\hat{m}^-_{st}(j)$; if $\hat{H}_{st}(i,j) > 0$, then $\hat{b}_{t\backslash s}(j)$ should be minimized, i.e., taking its lower bound $\hat{b}^-_t(j) - \frac{1}{k}\hat{m}^+_{st}(j)$. A similar idea applies to the computation of the upper bound of $\hat{m}_{ts}(i)$. Then by subtracting Eq. (16) from Eq. (17), we obtain the interval width $\mathbb{m}_{ts}(i)$ to reflect the uncertainty of $\hat{m}_{ts}(i)$:

$$\mathbb{m}_{ts}(i) \leftarrow k\sum_j |\hat{H}_{st}(i,j)|\left(\mathbb{b}_t(j) + \frac{1}{k}\mathbb{m}_{st}(j)\right), \tag{20}$$

where $\mathbb{b}_t$ is the interval width vector reflecting the posterior belief uncertainty at node $t$, as defined in the "Problem Definitions" of Sec. 2. Eq. (20) leaves only the interval width, making the specific location irrelevant for subsequent steps. Thus, *LinUProp* is also unaffected by the exact location of the interval. Similarly, the uncertainty of the message in the opposite direction can be quantified by interval width as:

$$\mathbb{m}_{st}(i) \leftarrow k\sum_j |\hat{H}_{st}(i,j)|\left(\mathbb{b}_s(j) + \frac{1}{k}\mathbb{m}_{ts}(j)\right). \tag{21}$$

Substituting Eq. (21) into Eq. (20) leads to

$$\mathbb{m}_{ts}(i) \leftarrow k \sum_j |\hat{H}_{st}(i,j)| \left(\mathbb{b}_t(j) + \sum_g |\hat{H}_{st}(g,j)| \left(\mathbb{b}_s(g) + \frac{1}{k}\mathbb{m}_{ts}(g)\right)\right). \qquad (22)$$

When the algorithm converges, all messages are at a stable state, so we can treat $\mathbb{m}_{ts}$ on both sides of Eq. (22) as equal and replace the update formula with the equation:

$$\mathbb{m}_{ts}(i) - \sum_j |\hat{H}_{st}(i,j)| \sum_g |\hat{H}_{st}(g,j)| \mathbb{m}_{ts}(g)$$
$$= k \sum_j |\hat{H}_{st}(i,j)| \mathbb{b}_t(j) + k \sum_j |\hat{H}_{st}(i,j)| \sum_g |\hat{H}_{st}(g,j)| \mathbb{b}_s(g). \qquad (23)$$

Let us denote $|H_{st}(i,j) - \frac{1}{k}|$ as $\hat{H}'_{st}(i,j)$, then the stable state message bound can be simplified as:

$$(\mathbf{I} - \hat{\mathbf{H}}'^2_{st})\mathbb{m}_{ts} = k\hat{\mathbf{H}}'_{st}\mathbb{b}_t + k\hat{\mathbf{H}}'^2_{st}\mathbb{b}_s$$
$$\mathbb{m}_{ts} = k(\mathbf{I} - \hat{\mathbf{H}}'^2_{st})^{-1}\hat{\mathbf{H}}'_{st}(\mathbb{b}_t + \hat{\mathbf{H}}'_{st}\mathbb{b}_s), \qquad (24)$$

where the message bound width, denoted as $\mathbb{m}_{ts}$, and the posterior belief bound width of nodes $s$ and $t$, denoted as $\mathbb{b}_s$ and $\mathbb{b}_t$ respectively, are all $k$-dimensional vectors. Then the interval width $\mathbb{b}_s$ reflecting the uncertainty of posterior belief $\hat{\mathbf{b}}_s$ can be calculated by subtracting its lower bound $\hat{\mathbf{b}}_s^- = \hat{\mathbf{e}}_s^- + \frac{1}{k}\sum_{t \in \mathcal{N}(s)} \hat{\mathbf{m}}_{ts}^-$ from its upper bound $\hat{\mathbf{b}}_s^+ = \hat{\mathbf{e}}_s^+ + \frac{1}{k}\sum_{t \in \mathcal{N}(s)} \hat{\mathbf{m}}_{ts}^+$:

$$\mathbb{b}_s = \mathbb{e}_s + \frac{1}{k}\sum_{t \in \mathcal{N}(s)} \mathbb{m}_{ts}, \qquad (25)$$

where $\mathbb{e}_s$ is the prior bound width vector with $k$-dimensional. Then plugging Eq. (24) into Eq. (25):

$$\mathbb{b}_s = \mathbb{e}_s + \sum_{t \in \mathcal{N}(s)} (\mathbf{I} - \hat{\mathbf{H}}'^2_{st})^{-1}\hat{\mathbf{H}}'_{st}\mathbb{b}_t + \sum_{t \in \mathcal{N}(s)} (\mathbf{I} - \hat{\mathbf{H}}'^2_{st})^{-1}\hat{\mathbf{H}}'^2_{st}\mathbb{b}_s.$$

Due to $\hat{\mathbf{H}}'_{st}$ being the centralized matrix, $(\mathbf{I} - \hat{\mathbf{H}}'^2_{st}) \approx \mathbf{I}$, then $(\mathbf{I} - \hat{\mathbf{H}}'^2_{st})^{-1}\hat{\mathbf{H}}'_{st} \approx \hat{\mathbf{H}}'_{st}$ and we get

$$\mathbb{b}_s = \mathbb{e}_s + \sum_{t \in \mathcal{N}(s)} \hat{\mathbf{H}}'_{st}\mathbb{b}_t + \sum_{t \in \mathcal{N}(s)} \hat{\mathbf{H}}'^2_{st}\mathbb{b}_s. \qquad (26)$$

By using the matrices introduced at the outset of this section, Eq. (26) can be rewritten as Eq. (6). $\quad\square$

## A.2   Proof of Eq. (13)

*Proof.* Based on the linear approximation of BP messages (Eq. (4)), we can similarly get $\hat{\mathbf{m}}_{st}$:

$$\hat{m}_{st}(i) \leftarrow k \sum_{j=1}^{k} \hat{H}_{st}(i,j) \left(\hat{b}_s(j) - \frac{1}{k}\hat{m}_{ts}(j)\right). \qquad (27)$$

Substituting Eq. (27) into Eq. (4) leads to

$$\hat{m}_{ts}(i) \leftarrow k \sum_j \hat{H}_{st}(i,j) \left(\hat{b}_t(j) - \sum_g \hat{H}_{st}(g,j) \left(\hat{b}_s(g) - \frac{1}{k}\hat{m}_{ts}(g)\right)\right). \qquad (28)$$

When the algorithm converges, all messages are at a stable state, so we can treat $\hat{\mathbf{m}}_{ts}$ on both sides of Eq. (28) as equal and replace the update formula with the equation:

$$\hat{m}_{ts}(i) - \sum_j \hat{H}_{st}(i,j) \sum_g \hat{H}_{st}(g,j) \hat{m}_{ts}(g)$$
$$= k \sum_j \hat{H}_{st}(i,j) \hat{b}_t(j) - k \sum_j \hat{H}_{st}(i,j) \sum_g \hat{H}_{st}(g,j) \hat{b}_s(g). \qquad (29)$$

This stable state message can be simplified as:

$$(\mathbf{I} - \hat{\mathbf{H}}^2_{st})\hat{\mathbf{m}}_{ts} = k\hat{\mathbf{H}}_{st}\hat{\mathbf{b}}_t - k\hat{\mathbf{H}}^2_{st}\hat{\mathbf{b}}_s$$
$$\hat{\mathbf{m}}_{ts} = k(\mathbf{I} - \hat{\mathbf{H}}^2_{st})^{-1}\hat{\mathbf{H}}_{st}(\hat{\mathbf{b}}_t - \hat{\mathbf{H}}_{st}\hat{\mathbf{b}}_s). \qquad (30)$$

Then plugging Eq. (30) into Eq. (5):

$$\hat{\mathbf{b}}_s = \hat{\mathbf{e}}_s + \sum_{t \in \mathcal{N}(s)} (\mathbf{I} - \hat{\mathbf{H}}_{st}^2)^{-1} \hat{\mathbf{H}}_{st} \hat{\mathbf{b}}_t + \sum_{t \in \mathcal{N}(s)} (\mathbf{I} - \hat{\mathbf{H}}_{st}^2)^{-1} \hat{\mathbf{H}}_{st} \hat{\mathbf{b}}_s.$$

Due to $\hat{\mathbf{H}}_{st}$ being the centralized matrix, $(\mathbf{I} - \hat{\mathbf{H}}_{st}^2) \approx \mathbf{I}$, then $(\mathbf{I} - \hat{\mathbf{H}}_{st}^2)^{-1} \hat{\mathbf{H}}_{st} \approx \hat{\mathbf{H}}_{st}$ and we get

$$\hat{\mathbf{b}}_s = \hat{\mathbf{e}}_s + \sum_{t \in \mathcal{N}(s)} \hat{\mathbf{H}}_{st} \hat{\mathbf{b}}_t + \sum_{t \in \mathcal{N}(s)} \hat{\mathbf{H}}_{st}^2 \hat{\mathbf{b}}_s. \tag{31}$$

By using $\mathbf{\Psi}_1$ and $\mathbf{\Psi}_2$ introduced at the outset of Sec. 4.2, Eq. (31) can be rewritten as Eq. (13).

$$\mathbf{\Psi}_1 = \begin{bmatrix} \hat{\mathbf{H}}_{11} & \cdots & \hat{\mathbf{H}}_{1n} \\ \vdots & \ddots & \vdots \\ \hat{\mathbf{H}}_{n1} & \cdots & \hat{\mathbf{H}}_{nn} \end{bmatrix} \quad \mathbf{\Psi}_2 = \begin{bmatrix} \hat{\mathbf{H}}_{11}^2 & \cdots & \hat{\mathbf{H}}_{1n}^2 \\ \vdots & \ddots & \vdots \\ \hat{\mathbf{H}}_{n1}^2 & \cdots & \hat{\mathbf{H}}_{nn}^2 \end{bmatrix}$$

$\square$

### A.3 Proof of the Linear Scalability of *LinUProp*

*Proof.* To demonstrate the linear scalability of *LinUProp*, we will start with the time complexity analysis of the iterative version of *LinUProp* (Eq. (7)):

$$\text{vec}(\mathbb{B})^{(l+1)} = \text{vec}(\mathbb{E}) + \left( \mathbf{\Psi}_1' + \text{Diag}\left( \mathbf{\Psi}_2' \mathbf{Q} \right) \right) \text{vec}(\mathbb{B})^{(l)}.$$

Before the iteration starts, $\text{vec}(\mathbb{E})$ and $\mathbf{\Psi}_1' + \text{Diag}(\mathbf{\Psi}_2' \mathbf{Q})$ (denoted as $\mathbf{T}$) are known and fixed as input. In the $l$-th iteration, we need to compute the matrix-vector multiplication $\mathbf{T} \cdot \text{vec}(\mathbb{B})^{(l)}$ and then add the result to $\text{vec}(\mathbb{E})$. Since $\mathbf{T}$ is a block sparse matrix, we can use sparse matrix operations to simplify the computation. The number of non-zero elements in $\mathbf{T}$ is $k^2 * (|\mathcal{E}| + |\mathcal{V}|)$, where $|\mathcal{E}|$ is the number of edges, $|\mathcal{V}|$ is the number of nodes and $k$ is the number of node classes. Then the time complexity of $\mathbf{T} \cdot \text{vec}(\mathbb{B})^{(l)}$ is $\mathcal{O}(k^2 * (|\mathcal{E}| + |\mathcal{V}|))$. Adding this result to $\text{vec}(\mathbb{E})$ has a complexity of $\mathcal{O}(k|\mathcal{V}|)$. Therefore, the time complexity of each iteration of *LinUProp* is $\mathcal{O}(k^2(|\mathcal{E}| + |\mathcal{V}|) + k|\mathcal{V}|)$. From this, we can conclude that the time complexity is $\mathcal{O}(|\mathcal{V}|)$ when $|\mathcal{V}| > |\mathcal{E}|$, otherwise $\mathcal{O}(|\mathcal{E}|)$, which reflects the linear scalability of *LinUProp*. $\square$

### A.4 Proof of Eq. 15

In order to expand the variance term, we first apply the property of variance, which states that the variance of a random variable is equal to the difference between the expected value of its square and the square of its expected value. Next, we substitute the linear function $\text{vec}(\hat{\mathbf{B}})_v = \mathbf{F}_v \text{vec}(\hat{\mathbf{E}})$ into the equation and appropriately expand the squared terms. Following this, we manipulate and simplify the terms using the properties of expectation and matrix operations. At this point, we factor out the linear function's coefficient vector $\mathbf{F}_v$. The resulting expression includes the covariance matrix of the random vector, denoted as $\Sigma_{\text{vec}(\hat{\mathbf{E}})}$. Finally, we present the concise form of the expression, which demonstrates the linear relationship between the variance term of each node and the prior covariances of all nodes. The detailed proof steps are provided below:

$$\mathbb{E}\left[ \left( \text{vec}(\hat{\mathbf{B}})_v - \mathbb{E}[\text{vec}(\hat{\mathbf{B}})_v] \right)^2 \right]$$

$$= \mathbb{E}\left[ \left( \text{vec}(\hat{\mathbf{B}})_v \right)^2 \right] - \left( \mathbb{E}\left[ \text{vec}(\hat{\mathbf{B}})_v \right] \right)^2$$

$$= \mathbb{E}\left[ \left( \mathbf{F}_v \text{vec}(\hat{\mathbf{E}}) \right)^2 \right] - \left( \mathbb{E}\left[ \mathbf{F}_v \text{vec}(\hat{\mathbf{E}}) \right] \right)^2$$

$$= \mathbb{E}\left[ \mathbf{F}_v \text{vec}(\hat{\mathbf{E}}) \text{vec}(\hat{\mathbf{E}})^T \mathbf{F}_v^T \right] - \mathbb{E}\left[ \mathbf{F}_v \text{vec}(\hat{\mathbf{E}}) \right] \left( \mathbb{E}\left[ \mathbf{F}_v \text{vec}(\hat{\mathbf{E}}) \right] \right)^T$$

$$= \mathbf{F}_v \mathbb{E}\left[ \text{vec}(\hat{\mathbf{E}}) \text{vec}(\hat{\mathbf{E}})^T \right] \mathbf{F}_v^T - \mathbf{F}_v \mathbb{E}\left[ \text{vec}(\hat{\mathbf{E}}) \right] \left( \mathbb{E}\left[ \text{vec}(\hat{\mathbf{E}}) \right] \right)^T \mathbf{F}_v^T$$

$$= \mathbf{F}_v \left( \mathbb{E}\left[ \text{vec}(\hat{\mathbf{E}}) \text{vec}(\hat{\mathbf{E}})^T \right] - \mathbb{E}\left[ \text{vec}(\hat{\mathbf{E}}) \right] \left( \mathbb{E}\left[ \text{vec}(\hat{\mathbf{E}}) \right] \right)^T \right) \mathbf{F}_v^T = \mathbf{F}_v \Sigma_{\text{vec}(\hat{\mathbf{E}})} \mathbf{F}_v^T$$

# B   Additional Details and Results

## B.1   Quantitative Validation of Correctness on a 4×4 Grid

We also quantitatively verify the correctness of *LinUProp* by comparing it with the uncertainty computed by MC simulations on a $4 \times 4$ grid. Specifically, we first sample prior beliefs based on the prior distributions of each node, and then run BP to infer the posterior beliefs of each node. After repeating the above sampling process for 100,000 times, we use the empirical standard deviation of the posterior belief of each node sampled as an estimation of the ground truth uncertainty.

Figure 7 shows that the uncertainties quantified by MC simulation and *LinUProp* have a strong positive correlation, as evidenced by a Pearson Correlation Coefficient (PCC) of 0.9084 between the uncertainties obtained for all nodes using both approaches. This confirms the consistency of the uncertainty in the beliefs quantified by our method with that of MC simulation.

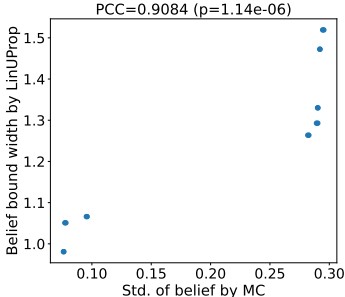

Figure 7: Correctness. The two axes represent two UQ methods: the x-axis is the empirical standard deviation of the beliefs obtained by MC simulation (an estimation of the ground truth uncertainty), and the y-axis is the belief bound width computed by *LinUProp*. In fact, there are 16 points in this figure, corresponding to the 16 variable nodes in Fig. 3(a). Some points appear to overlap because they have similar uncertainties quantified by both methods. The Pearson Correlation Coefficient (PCC) between the uncertainties quantified by the two methods is 0.9084, indicating a strong positive correlation.

## B.2   More Experimental Details

**Running environment.** We conducted convergence and scalability experiments on the Apple M2 chip, and convergence experiments on a 2.2 GHz Intel Xeon CPU.

**Datasets.** Table 1 shows the statistical information and partitioning of three citation network datasets.

Table 1: Statistical information and partitioning of datasets. The subsets, $\mathcal{V}_{train}$, $\mathcal{V}_{val}$ and $\mathcal{V}_{test}$ are sampled from the original node set. The remaining nodes are in $\mathcal{V}_{ulp}$. We use these subsets in the effectiveness experiments.

| Dataset | #Nodes | #Edges | #Classes | #$\mathcal{V}_{train}$ | #$\mathcal{V}_{val}$ | #$\mathcal{V}_{test}$ |
|---|---|---|---|---|---|---|
| Cora | 2,708 | 5,429 | 7 | 14 | 500 | 1,000 |
| Citeseer | 3,327 | 4,732 | 6 | 12 | 500 | 1,000 |
| PubMed | 19,717 | 44,338 | 3 | 6 | 500 | 1,000 |
| PolBlogs | 1,490 | 19,090 | 2 | 4 | 250 | 500 |

## B.3 More Effectiveness Results for *LinUProp*

### B.3.1 Results under Varying Labeling Budgets with More Labeling Accuracy Settings

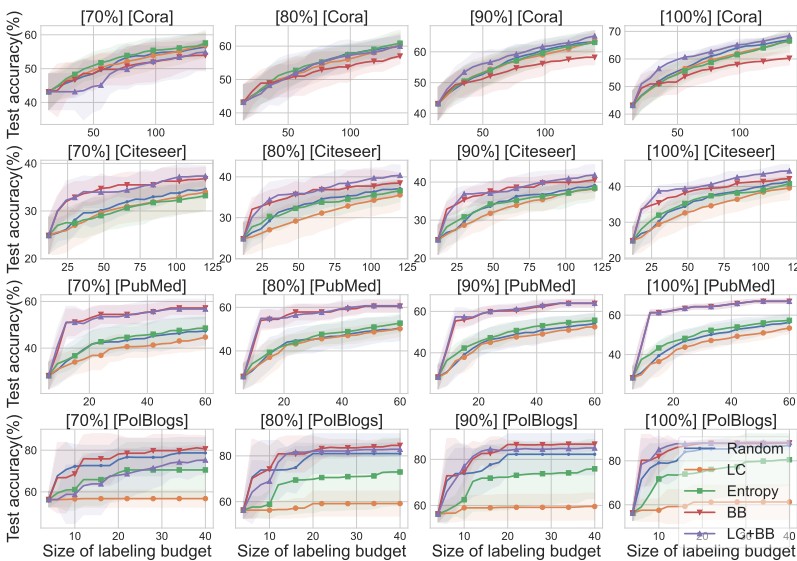

Figure 8: Test accuracy for varying labeling budgets with **BP** inferring posterior beliefs using noisy labeled nodes. Each subplot title includes two components, which represent the labeling accuracy and dataset. Each row corresponds to a dataset, and each column corresponds to a labeling accuracy. In each subplot, the node selection strategies based on *LinUProp* and its variant are represented by red ▼ and purple ▲. Under the same labeling budget, the higher the test accuracy, the better.

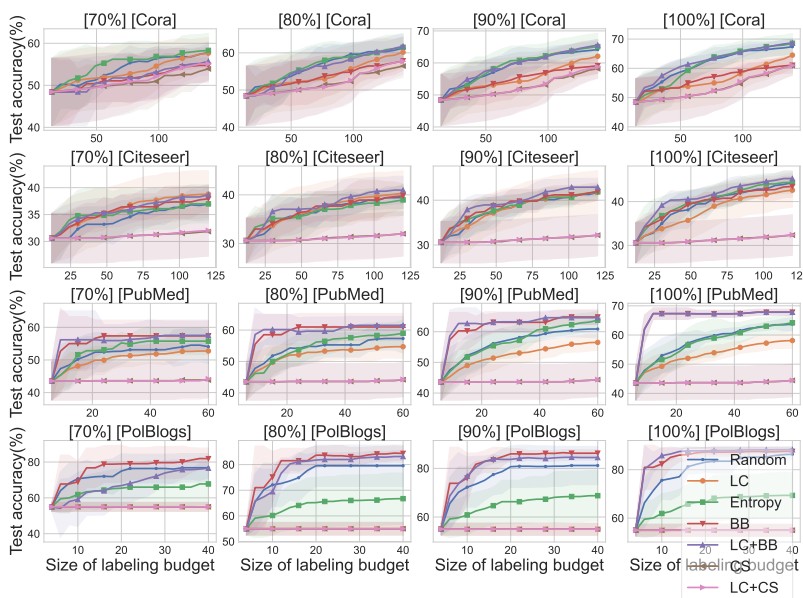

Figure 9: Test accuracy for varying labeling budgets with **NETCONF** inferring posterior beliefs using noisy labeled nodes. Each subplot title includes two components, which represent the labeling accuracy and dataset. Each row corresponds to a dataset, and each column corresponds to a labeling accuracy. In each subplot, the node selection strategies based on *LinUProp* and its variants are represented by red ▼ and purple ▲, while the strategies based on the native UQ method in NETCONF and its variants are represented by brown ◄ and pink ►. NETCONF-based strategies (CS/LC+CS) often prioritize labeling low-degree nodes due to their assumption that neighbors serve as evidence, leading to persistent high uncertainty in most nodes and slowly increasing accuracy. Under the same labeling budget, the higher the test accuracy, the better.

## B.3.2 Results under a fixed Labeling Budgets

Tables 2 and 3 show the mean test accuracies and standard deviations for a fixed labeling budget (20$b$) with BP/NETCONF inference on noisy labeled nodes. *LinUProp* achieves the highest mean accuracy in 28 of 32 cases, consistently outperforming its competitors across all four datasets, two inference methods, and four labeling accuracies. While the performance improvements are not always statistically significant, *LinUProp* maintains lower standard deviations (below 7%) and avoids poor accuracy. In contrast, LC achieves only 55% accuracy on Polblogs in Table 3, while *LinUProp* consistently exceeds 80%. The Entropy method shows high standard deviations (up to 14%) on Polblogs, whereas *LinUProp* generally has more stable performance, as underlined in the results.

Table 2: Mean test accuracies and their standard deviations for a fixed labeling budget (20$b$) with **BP** inferring posterior beliefs using noisy labeled nodes. Each cell shows the mean accuracy and its standard deviation on the test set for different labeling accuracy and node selection strategies, on different datasets. BB and LC+BB are node selection strategies based on *LinUProp*. **Bold** values indicate the highest mean accuracy. Underlined values emphasize the method with the lower standard deviation between the *LinUProp* winner and the non-*LinUProp* winner. Superscripts indicate significant superiority between the *LinUProp* winner and the non-*LinUProp* winner (pairwise t-test at a 5% significance level (*), 10% significance level (†)).

| Dataset | | Random | LC | Entropy | BB | LC+BB |
|---|---|---|---|---|---|---|
| Cora | 100% | 66.900±1.987 | 66.450±3.188 | 66.590±3.853 | 60.270±2.451 | **68.380±1.130***  |
| | 90% | 63.280±2.665 | 63.560±3.463 | 63.060±3.640 | 58.230±2.450 | **65.140±1.748†** |
| | 80% | 60.130±3.031 | 60.230±3.764 | **60.880±3.894** | 57.010±2.221 | 59.970±3.378 |
| | 70% | 56.400±2.352 | 56.950±3.798 | **57.570±3.793** | 53.910±4.266 | 54.880±5.502 |
| Citeseer | 100% | 41.450±1.370 | 39.580±3.261 | 40.790±1.957 | 42.180±2.144 | **44.360±1.737*** |
| | 90% | 39.120±1.584 | 38.200±2.279 | 38.420±3.069 | 40.570±1.521 | **41.860±2.689*** |
| | 80% | 37.130±1.920 | 35.560±3.482 | 36.610±2.678 | 38.470±2.410 | **40.360±2.516*** |
| | 70% | 34.600±1.849 | 33.990±3.734 | 33.220±3.160 | 36.800±2.426 | **37.340±2.183*** |
| Pubmed | 100% | 56.360±4.270 | 53.440±3.001 | 57.350±1.827 | **67.030±1.057*** | **67.030±1.057*** |
| | 90% | 54.070±5.053 | 52.530±4.403 | 55.730±3.071 | 63.900±1.825 | **63.940±1.943*** |
| | 80% | 50.550±5.762 | 50.180±3.624 | 52.750±4.055 | 60.560±2.696 | **60.740±2.996*** |
| | 70% | 47.350±6.072 | 44.740±5.398 | 48.580±4.519 | **57.210±2.763*** | 56.760±4.201 |
| PolBlogs | 100% | 86.920±1.847 | 61.280±7.838 | 80.520±11.685 | 88.140±1.051 | **88.340±1.230*** |
| | 90% | 82.200±8.908 | 59.540±5.826 | 75.880±12.847 | **86.600±0.934†** | 84.920±1.443 |
| | 80% | 80.980±8.674 | 59.180±5.475 | 72.960±14.731 | **84.560±2.874** | 82.880±2.685 |
| | 70% | 78.700±7.382 | 56.740±4.103 | 70.520±13.114 | **80.620±6.167** | 75.280±6.752 |

Table 3: Mean test accuracies and their standard deviations for a fixed labeling budget (20$b$) with **NETCONF** inferring posterior beliefs using noisy labeled nodes. Each cell shows the mean accuracy and its standard deviation on the test set for different labeling accuracy and node selection strategies, on different datasets. BB and LC+BB are node selection strategies based on *LinUProp*. CS and LC+CS are node selection strategies based on the native UQ methods of NETCONF. **Bold** values indicate the highest mean accuracy. Underlined values emphasize the method with the lower standard deviation between the *LinUProp* winner and the non-*LinUProp* winner. Superscripts indicate significant superiority between the *LinUProp* winner and the non-*LinUProp* winner (pairwise t-test at a 5% significance level (*), 10% significance level (†)).

| Dataset | | Random | LC | Entropy | BB | LC+BB | CS | LC+CS |
|---|---|---|---|---|---|---|---|---|
| Cora | 100% | 67.380±2.458 | 64.520±4.241 | 68.570±3.373 | 61.060±2.621 | **68.670±1.311** | 60.950±3.300 | 61.000±3.305 |
| | 90% | 64.150±2.727 | 62.090±5.278 | 65.110±4.112 | 59.130±2.384 | **65.580±1.736** | 58.170±2.981 | 58.770±3.515 |
| | 80% | 61.260±3.017 | 60.160±4.021 | 61.500±3.731 | 57.850±2.570 | **61.770±3.223** | 56.450±2.987 | 57.730±3.473 |
| | 70% | 57.810±2.089 | 57.660±3.696 | **58.310±4.051†** | 54.740±4.113 | 55.720±4.635 | 53.910±2.542 | 55.080±2.740 |
| Citeseer | 100% | 44.340±1.322 | 42.600±3.907 | 44.340±2.480 | 43.440±2.406 | **45.250±1.806** | 32.360±4.584 | 32.400±4.658 |
| | 90% | 41.740±1.796 | 41.860±4.742 | 41.860±1.489 | 41.950±1.980 | **42.910±1.980** | 32.190±4.598 | 32.200±4.635 |
| | 80% | 39.460±2.398 | 40.370±4.609 | 38.960±3.186 | 39.810±2.461 | **40.980±2.813** | 32.000±4.556 | 31.990±4.636 |
| | 70% | 36.950±2.916 | **38.800±4.364** | 36.980±3.543 | 37.940±2.441 | 38.690±1.842 | 31.840±4.595 | 32.020±4.725 |
| Pubmed | 100% | 63.730±2.287 | 58.110±2.653 | 64.180±2.755 | **67.820±1.036*** | **67.820±1.036*** | 44.450±5.637 | 44.450±5.637 |
| | 90% | 60.870±2.950 | 56.520±2.546 | 63.710±2.926 | **64.790±2.438** | 64.560±2.229 | 44.350±5.547 | 44.380±5.627 |
| | 80% | 57.310±4.673 | 54.700±3.656 | 58.980±4.245 | 60.980±4.326 | **61.520±2.747†** | 44.190±5.397 | 44.270±5.498 |
| | 70% | 54.070±5.093 | 52.760±2.634 | 55.770±4.171 | 57.380±4.829 | **57.540±4.408** | 43.850±5.528 | 44.190±5.592 |
| PolBlogs | 100% | 86.360±2.095 | 55.000±2.475 | 69.380±9.463 | 87.560±1.397 | **88.240±1.311*** | 55.000±2.475 | 55.000±2.475 |
| | 90% | 81.180±7.845 | 55.000±2.475 | 68.780±10.411 | **86.240±1.402†** | 84.460±1.103 | 55.000±2.475 | 55.000±2.475 |
| | 80% | 79.540±8.048 | 55.000±2.475 | 66.720±9.959 | **84.360±2.707†** | 83.140±2.265 | 55.000±2.475 | 55.000±2.475 |
| | 70% | 76.800±7.949 | 55.000±2.475 | 67.760±11.368 | **81.880±5.860†** | 76.540±5.536 | 55.000±2.475 | 55.000±2.475 |

## B.4 Additional Runtime Results

Under the same experimental conditions as the linear scalability experiments in Figure 4(b), we tested NETCONF on four datasets, comparing its computation time (including all edges) with that of *LinUProp*. As shown in Table 4, the computation times of NETCONF and *LinUProp* are quite similar. However, *LinUProp* demonstrates a significant performance advantage over the NETCONF-based UQ method, as evident from Table 3 (BB/LC+BB vs. CS/LC+CS).

Table 4: Runtime comparison including all edges across different datasets (in seconds)

| Method | Cora | Citeseer | Pubmed | Polblogs |
|---|---|---|---|---|
| **NETCONF** | 0.0225 | 0.0216 | 0.0627 | 0.0126 |
| *LinUProp* | 0.0256 | 0.0181 | 0.0559 | 0.0084 |

