# OpenReview forum: "Linear Uncertainty Quantification of Graphical Model Inference"
_NeurIPS.cc/2024/Conference — NeurIPS 2024 poster_

### Official Review · Reviewer_G7cz · 2024-07-06

**Soundness:** 2
**Presentation:** 3
**Contribution:** 3
**Rating:** 6
**Confidence:** 3

**Summary:**

This paper proposes an uncertainty quantification method for graphical models that uses linear propagation to model uncertainty. Experiments show that it can achieve competitive results and fast convergence in downstream tasks.

**Strengths:**

1: The paper is well-written. The motivation for the paper and the proposed solutions are clear, making it easy for readers to follow.

2: Theoretical analysis is solid. The authors clearly explain the necessary preliminaries and provide detailed proofs.

**Weaknesses:**

1: Most of the datasets used for validating the properties of the proposed method are datasets with high homophily, which makes the validation less comprehensive. It would be better to conduct experiments on several datasets with high heterophily.

2: The illustrations in some of the figures are vague. For example, Figure 1 is quite inclusive but lacks some necessary explanations. The meaning of distribution Beta and the values of alpha and beta are hard to understand (though I clearly understand the calculation part).

3: The formatting of the paper is less professional, especially in the appendix. In B.3.2, there are large blanks between the title and Table 2. Also, Table 3 is oversized.

**Questions:**

Can you please explain the upper part of Figure 2? I'm a bit confused about its meaning.

---

> ### Author Rebuttal · Authors · 2024-08-07
>
> Thanks for your detailed and constructive reviews.
>
> ## Regarding the Formatting Issues
>
> >The formatting of the paper is less professional, especially in the appendix. In B.3.2, there are large blanks between the title and Table 2. Also, Table 3 is oversized.
>
> Thank you for pointing out these formatting issues. We will carefully review and address all formatting concerns, especially in the appendix, and make the necessary corrections in the camera-ready version.
>
> ## Further Explanation of the Figures
>
> >The illustrations in some of the figures are vague. For example, Figure 1 is quite inclusive but lacks some necessary explanations. The meaning of distribution Beta and the values of alpha and beta are hard to understand (though I clearly understand the calculation part).
>
> Let's understand the meaning of the Beta distribution and its parameters in Figure 1 within the context of an active learning scenario. Consider a graph active learning task where each annotator can label a node as one of two categories. In this case, the Beta distribution can be used to model the uncertainty in the probability of a node belonging to a specific category. When considering a node individually,  $\alpha$ can be interpreted as the number of times the node is labeled as category A plus one, and $\beta$ can be interpreted as the number of times the node is labeled as category B plus one. The sum $\alpha + \beta$ represents the total number of times the node has been labeled plus two; the larger this sum, the more times the node has been labeled, and thus the lower the uncertainty.
>
> Thank you for highlighting the points that might confuse readers. We will address this in the camera-ready version by incorporating the specific scenarios mentioned above to make Figure 1 easier to understand.
>
> >Can you please explain the upper part of Figure 2? I'm a bit confused about its meaning.
>
> The upper part of Figure 2 illustrates, through a 3-node chain graphical model, what information we need to input if we want to quantify the uncertainty in the posterior belief of each node using LinUProp. To make this easier to understand, we can think of it as a social network.
>
> In a simple social network with 3 users, each user can be categorized into one of two groups based on whether they are music enthusiasts or not. If we want to know the uncertainty in the posterior probability of each user being a music enthusiast, considering the influence of other users, we need to provide two types of information:
>
> - The interval width representing the uncertainty about whether each user is a music enthusiast when considered individually (i.e., the prior interval widths $\mathbb{e}_1, \mathbb{e}_2, \mathbb{e}_3$). This could be derived from user-filled profiles. If a user hasn't specified their music preferences, we assign a wider interval width; otherwise, we assign a narrower interval width.
> - The "compatibility matrix" between every two connected users. For example,
> $$ \mathbf{H}_{12}=\begin{bmatrix} 0.8 & 0.2 \\\\ 0.2 & 0.8 \end{bmatrix} $$
>
> $\mathbf{H}_{12}(i,j)$ denotes the degree of association between class i of user 1 and class j of user 2.
> This matrix suggests that users 1 and 2 are more likely to either both like or both dislike music, rather than only one of them liking music.
>
> >Most of the datasets used for validating the properties of the proposed method are datasets with high homophily, which makes the validation less comprehensive. It would be better to conduct experiments on several datasets with high heterophily.
>
> LinUProp is a UQ method, so we expect to experimentally verify its performance in uncertainty quantification. We evaluate UQ methods in graph active learning tasks through an uncertainty-based node selection strategy because a good UQ method can prioritize nodes with high uncertainty for labeling, achieving the highest possible accuracy within a limited labeling budget. This approach is reasonable for validating UQ methods on homophily graphs, as prioritizing high-uncertainty nodes can help clarify the categories of neighboring nodes, and after propagation, quickly reduce global uncertainty, achieving higher accuracy with a lower labeling budget. However, on heterophily graphs, even if high-uncertainty nodes are prioritized for labeling, it may not help clarify the categories of neighboring nodes (especially when there are many node categories). In such cases, prioritizing high-uncertainty nodes may not lead to a rapid increase in accuracy, thus failing to appropriately reflect the performance of the UQ method.

---

> > ### Comment · Reviewer_G7cz · 2024-08-09
> > **Response to Author‘s Rebuttal**
> >
> > I would like to thank the authors for their detailed discussion and for addressing my questions. The explaination of the figures is essential for me to understand the paper. This is a good work. I sincerely hope the authors can further improve the paper to make it better. I've increased my score. Good luck!

---

> > > ### Author Response · Authors · 2024-08-10
> > >
> > > We appreciate your informative feedback, which is extremely useful to make our work better. We are also glad that our explanations are helpful. Thank you for your efforts and support!

---

### Official Review · Reviewer_rPk6 · 2024-07-12

**Soundness:** 3
**Presentation:** 3
**Contribution:** 3
**Rating:** 7
**Confidence:** 3

**Summary:**

[Edit: After discussion, the Authors have addressed my concerns on the presentation and discussion of their results. I have accordingly increased my score (previously 5).]

The paper proposes an alternative algorithm for message passing for uncertainty quantification in graphical models. The method is linear, with (presumed) computational performance gains over existing methods, less susceptibility to certain kinds of bias, and supporting theoretical justification. The method is empirically explored in simulation studies.

**Strengths:**

Theoretical justification of the method is helpful.

**Weaknesses:**

Some areas were hard to understand. E.g. the motivating discussion of why some (but not all) existing methods could only reduce uncertainty.

Limitations of the method have not really been addressed. It seems the proposed method is perfect!

**Questions:**

I was a little confused on the comparative studies.

1) It isn't clear to me whether the proposed method produces unbiased results (though my guess is that it does not, and I note the content in section 4.2 to this end, and the claim it avoids bias in Section 7). Other methods, as discussed, do provide unbiased results, but perhaps they are very computational or have other downsides. So its ok for the proposed method to be biased if there is a substantial speed gain, for example. (*Is* there a speed gain? Actual comparisons with other methods don't seem to have been made.)  Or, put another way, the method presumably gains lots by designing for linear speed. What did it lose?

2) I couldn't find anywhere which described what "BP" or "NETCONF" were (Figures 5-6). This somewhat reduces the readers ability to understand the information being presented.

3) Tables 2 and 3 (Appendix) have generously bolded the results corresponding to the papers methodology, but the +/- standard errors of the means presented strongly suggest that in many cases there is no statistically significant difference between many of the methods (bolded, or otherwise). This would seem to undermine the accuracy performance claims.

**Limitations:**

The paper checklist states that the limitations are outlined in the Conclusion (section 7). However, no limitations of the proposed methods seem to be discussed here. (The statement "However, ..." is not a limitation.) I would expect that the authors really ought to have some things to say here. (Presumed) strong computational gains of methods usually have a tradeoff.

---

> ### Author Rebuttal · Authors · 2024-08-07
>
> Thanks for your detailed and constructive reviews.
>
> >Some areas were hard to understand. E.g. the motivating discussion of why some (but not all) existing methods could only reduce uncertainty.
>
> The sentence you mentioned in the original text is: “However, this results in **any neighbor of a node will necessarily reduce the uncertainty**, even neighbors with noise or missing information, thus underestimate posterior uncertainty, as depicted in Figure 1.” In the previous sentence, we explained the reason as “Existing works [7, 35]...modeling beliefs as Dirichlet distributions and **treating neighboring nodes as observations**.” At the end of this sentence, we indicated that Figure 1 provides a further explanation of this point. Existing methods [7, 35] treat neighbors as **"observations,"** implicitly assuming that a node’s neighbors **are always evidence** and thus will necessarily reduce uncertainty.
>
> >Limitations of the method have not really been addressed. It seems the proposed method is perfect!
> >The paper checklist states that the limitations are outlined in the Conclusion (section 7). However, no limitations of the proposed methods seem to be discussed here. (The statement "However, ..." is not a limitation.) I would expect that the authors really ought to have some things to say here. (Presumed) strong computational gains of methods usually have a tradeoff.
>
> From the derived convergence conditions of LinUProp ($\rho(\mathbf{\Psi_{1}}^{'}+\text{Diag}(\mathbf{\Psi_{2}}^{'}\mathbf{Q}))<1$), we can see that when the graph is very large, if there is strong global (most edges) homophily/heterophily, the norm of the matrix  $T=\mathbf{\Psi_{1}}^{'}+\text{Diag}(\mathbf{\Psi_{2}}^{'}\mathbf{Q})$ will be large, which may cause LinUProp to fail to satisfy the convergence condition of the spectral radius of $T$ being less than 1. However, if only local (a few edges) strong homophily/heterophily exists in a large-scale graph, LinUProp can still converge. This is a limitation of LinUProp, and we will include the above discussion in the camera-ready version.
>
>
> >It isn't clear to me whether the proposed method produces unbiased results ......
>
>  The term "unbiased" in the abstract was intended to communicate that our method aims to avoid the underestimation of uncertainty, which some other methods might suffer from (NETCONF/SocNL). We didn't mean to claim that our method is mathematically unbiased in the statistical sense without further rigorous proof, although we validated the consistency of LinUProp with MC simulation through the Correctness experiment in Section 5.1. This issue appears only in two sentences in the abstract and one sentence in the conclusion, and is not mentioned elsewhere in the original text. To avoid any confusion, we propose to revise the two sentences in the abstract and the one sentence in the conclusion. Additionally, we have not yet studied the relationship between the speed of LinUProp and its unbiasedness, which might be a worthwhile topic for future research.
>
>  - Abstract
> 	- There are fast UQ methods for graphical models with closed-form solutions and convergence guarantee but with ~~biased~~ uncertainty underestimation.
> 	- We propose LinUProp, a UQ method that utilizes a
> novel linear propagation of uncertainty to model uncertainty among related nodes additively instead of multiplicatively, to offer linear scalability, guaranteed convergence, and ~~unbiased~~ closed-form solutions that do not underestimate uncertainty.
> - Conclusion
> 	- Unlike its competitors, LinUProp does not assume neighbors necessarily reduce uncertainty and thus avoids ~~biased~~ uncertainty underestimation.
>
> >I couldn't find anywhere which described what "BP" or "NETCONF" were (Figures 5-6). This somewhat reduces the readers ability to understand the information being presented.
>
> As stated in the captions of Figures 5 and 6, BP and NETCONF are two methods for **inferring posterior beliefs**. We have detailed Belief Propagation (BP) in the Preliminaries and introduced how NETCONF works through a simple case in Figure 1. The key difference between the two is that BP can only provide a point estimate of the posterior beliefs, whereas NETCONF can derive a Dirichlet posterior from which a "Certainty Score" can be obtained. This "Certainty Score" can be used to estimate the uncertainty of the posterior, and its calculation method is provided in lines 275-277.
>
> >Tables 2 and 3 (Appendix) have generously bolded the results corresponding to the papers methodology, but the +/- standard errors of the means presented strongly suggest that in many cases there is no statistically significant difference between many of the methods (bolded, or otherwise). This would seem to undermine the accuracy performance claims.
>
> Thank you for your question regarding standard deviations and statistical significance. We have added annotations addressing these two points to Tables 2 and 3 in the global response PDF. Specifically, we have used $\underline{\text{underlined}}$ values to emphasize the method with the lower standard deviation between the LinUProp winner and the non-LinUProp winner. We have also used superscripts to indicate significant superiority between the LinUProp winner and the non-LinUProp winner (pairwise t-test at a 5% significance level (*) and 10% significance level ($\dagger$)). We can observe that the LinUProp-based methods (BB/LC+BB) consistently maintain low standard deviations across 4 datasets, 2 inference methods, and 4 labeling accuracies. This level of consistency is not exhibited by other methods, even though some approaches may achieve comparable accuracy to LinUProp in certain specific cases.

---

> > ### Comment · Reviewer_rPk6 · 2024-08-12
> >
> > Thank you to the authors for the detailed responses.
> >
> > * Some areas were hard to understand.
> >
> > Thank you for the explanation. Of course, in the paper I had read all of these parts (several times) and still struggled to understand (merely saying something doesn't make it understandable), and I see another reviewer had similar issues. Information density is one part of this. It would be good if the explanations in this area were given a once over to perhaps improve the explanation in case others have similar issues. Similarly for the visibility of BP/NETCONF - if one has to look into a densely written figure caption to find the reference to support a method mentioned in another figure caption ... something is wrong with the presentation.
> >
> > * Limitations
> >
> > Thank you - limitations should always be clearly stated and not hidden. As a result, I do think that some mention needs to be given in the current paper for the computational overheads of LinUProp for the simulations run. If the method takes 10 times longer to run than the competitor and only achieves a small improvement, this is information that should be provided to the reader, right?
> >
> > * Statistical significance in Tables 2 & 3
> >
> > Thank you for the modified tables - this is much more informative than before. It now offers at-a-glance understanding of where there is evidence that LinUProp actually does beat the best of the competitors. 10% significance is probably a stretch though. So now it's notable that there is only a small amount evidence for improved performance over BP for PolBlogs (Table 2) and Cora, and not much at all over NETCONF for almost all datasets. I hope that this information will be fairly discussed in Section 5, as there is *some* evidence of improved performance in *some* cases, but the overall evidence here isn't strong.
> >
> > I note that you have control over the sample size (here n=10) for this t-test, so you could potentially improve these outcomes with improved numbers of simulations (which should have been done in the first place; hint). I also note that its ok to get exactly the same performance as a competitor method if you're doing the analysis much faster (see above point). Finally, its also ok to get the same performance as a competitor for exactly the same computational overheads if there is conceptually some other advantage to be had, but this case is the authors to make.
> >
> > * Thank you for your other responses (no further comments).

---

> > > ### Author Response · Authors · 2024-08-12
> > > **Further Response to Reviewer rPk6 (2)**
> > >
> > > > -   Limitations
> > > Thank you - limitations should always be clearly stated and not hidden. As a result, I do think that some mention needs to be given in the current paper for the computational overheads of LinUProp for the simulations run. If the method takes 10 times longer to run than the competitor and only achieves a small improvement, this is information that should be provided to the reader, right?
> > >
> > > The computational cost of LinUProp has been  **explicitly**  presented in Figure 4(b). Figure 4(b) demonstrates LinUProp's linear scalability, where the last data point of each dataset's corresponding line represents LinUProp's runtime when including all edges in that dataset. To address your concerns about LinUProp's computational cost relative to competitors, we will further discuss the comparison of computation times between LinUProp and its competitors.
> > >
> > > Under the same experimental environment as the linear scalability experiments above, we first tested NETCONF on 4 datasets, comparing its computation time when including all edges with that of LinUProp. The following table shows the runtime (in seconds):
> > >
> > > | Method   | Cora   | Citeseer | Pubmed | Polblogs |
> > > |----------|--------|----------|--------|----------|
> > > | NETCONF  | 0.0225 | 0.0216   | 0.0627 | 0.0126   |
> > > | LinUProp | 0.0256 | 0.0181   | 0.0559 | 0.0084   |
> > >
> > > From the table above, it's evident that the computation times of NETCONF and LinUProp are similar. However, LinUProp's performance shows a significant advantage over the NETCONF-based UQ method, which can be seen at a glance from Tables 2 and 3 (BB/LC+BB vs CS/LC+CS).
> > >
> > > Regarding the unbiased MC simulation method, it is highly time-consuming and therefore difficult to use as an algorithm for selecting nodes in active learning. In our experiments in Section 5.1, we verified the consistency between LinUProp and MC (100,000 samples) on small graphs. Below are the times (in seconds) required for **just one** sampling on each dataset (in seconds):
> > > | Cora  | Citeseer | Pubmed  | Polblogs |
> > > |-------|----------|---------|----------|
> > > | 4.4634 | 4.2388  | 52.5918 | 9.8378   |
> > >
> > > For the "Least Confidence" and "Entropy" methods, they only require simple calculations of confidence and entropy for each node, respectively. However, calculating the confidence or entropy for each node requires first computing the posterior probability for each node. If BP is used as the inference method, obtaining the posterior probability for each node requires running BP once, which takes as long as performing one MC sampling in the table above, and is **significantly slower than LinUProp**. If NETCONF is used as the inference method, then its runtime will be similar to that of LinUProp. We will clarify LinUProp's unique advantages compared to these two methods in our response to the next question.

---

> > > > ### Comment · Reviewer_rPk6 · 2024-08-12
> > > >
> > > > Thank you for providing the relative time comparisons for the competitor models (I acknowledge the absolute times for LinUProp in Fig 4,b). These give a greater grounding and believability for the results and simulations present. Extremely helpful.

---

> ### Author Response · Authors · 2024-08-12
> **Further Response to Reviewer rPk6 (1)**
>
> Thank you for your feedback. Regarding the concerns you raised, it's our pleasure to provide further clarification:
>
> >-   Some areas were hard to understand.
> Thank you for the explanation. Of course, in the paper I had read all of these parts (several times) and still struggled to understand (merely saying something doesn't make it understandable), and I see another reviewer had similar issues. Information density is one part of this. It would be good if the explanations in this area were given a once over to perhaps improve the explanation in case others have similar issues.
>
> Thank you for helping us understand that the following description alone might still confuse some readers: "Existing methods [7, 35] treat neighbors as  **"observations,"**  implicitly assuming that a node's neighbors  **are always evidence**  and thus will necessarily reduce uncertainty."
>
> A fundamental assumption of existing works [7, 35] (NETCONF/SocNL) is that "a node's neighbors are always evidence." Let's consider a simple social network example where the task is to determine whether each user is a music enthusiast. NETCONF/SocNL use the Beta distribution ($\mathcal{B}(\alpha,\beta)$) to describe the uncertainty of class probabilities in binary classification tasks, where parameters $\alpha$ and $\beta$ are pseudo-counts. In this context, $\alpha$ and $\beta$ can be understood as virtual number of times a user has been observed to like or dislike music, thus a larger $\alpha+\beta$ indicates lower uncertainty. Let's assume that a user who claimed to be a music enthusiast in their profile follows $\mathcal{B}(10,1)$, one who claimed not to be follows $\mathcal{B}(1,10)$, and one who didn't provide this information follows $\mathcal{B}(1,1)$ (equivalent to a uniform distribution over the interval [0, 1]).
>
>
> User A hasn't provided information about being a music enthusiast$(\mathcal{B}(1,1))$, but has two friends who are($\mathcal{B}(10,1)$). User B also hasn't provided this hobby information$(\mathcal{B}(1,1))$, and in addition to having two music enthusiast friends($\mathcal{B}(10,1)$), also has two friends who haven't specified their hobby information$(\mathcal{B}(1,1))$.
>
> NETCONF/SocNL, based on the assumption that "neighbors are always evidence," will use neighbors as evidence to update A and B. **This is achieved by adding the parameters of neighbors to one's own.** Under this assumption, any neighbor will inevitably increase $\alpha+\beta$, thereby leading to lower uncertainty after the update. Returning to the above example, after updating, A will follow $\mathcal{B}(1+10+10,1+1+1)=\mathcal{B}(21,3)$, while B will follow $\mathcal{B}(1+10+10+1+1,1+1+1+1+1)=\mathcal{B}(23,5)$. At this point, although both A and B would be considered music enthusiasts, B's uncertainty would be lower because B has more evidence (neighbors). Despite the two additional neighbors not providing information, B would still be considered to have lower uncertainty. Consequently, NETCONF/SocNL would be more confident in considering B a music enthusiast.
>
>
>
>
>
>
> > Similarly for the visibility of BP/NETCONF - if one has to look into a densely written figure caption to find the reference to support a method mentioned in another figure caption ... something is wrong with the presentation.
>
> Thank you for your valuable feedback. To make it easier for readers to find the introduction to NETCONF without having to look at the caption of another figure, we will explicitly include the introduction to NETCONF in the Preliminary section. Currently, there is already an introduction to BP in the Preliminary section. After the improvement, BP and NETCONF will be presented together, enhancing readability.

---

> > ### Comment · Reviewer_rPk6 · 2024-08-12
> >
> > Thank you - this expanded discussion/example is *so* much clearer. The extra space that this will take up in the paper will be very much worth it.

---

> ### Author Response · Authors · 2024-08-12
> **Further Response to Reviewer rPk6 (3)**
>
> >-   Statistical significance in Tables 2 & 3
> Thank you for the modified tables - this is much more informative than before. It now offers at-a-glance understanding of where there is evidence that LinUProp actually does beat the best of the competitors. 10% significance is probably a stretch though. So now it's notable that there is only a small amount evidence for improved performance over BP for PolBlogs (Table 2) and Cora, and not much at all over NETCONF for almost all datasets. I hope that this information will be fairly discussed in Section 5, as there is  _some_  evidence of improved performance in  _some_  cases, but the overall evidence here isn't strong.
> I note that you have control over the sample size (here n=10) for this t-test, so you could potentially improve these outcomes with improved numbers of simulations (which should have been done in the first place; hint). I also note that its ok to get exactly the same performance as a competitor method if you're doing the analysis much faster (see above point). Finally, its also ok to get the same performance as a competitor for exactly the same computational overheads if there is conceptually some other advantage to be had, but this case is the authors to make.
>
> Thank you for your further questions. We will now proceed with a more in-depth analysis of Tables 2 & 3, as well as discuss the advantages of LinUProp compared to its competitors.
>
> - From the perspective of runtime, as we replied in the previous question:
> 	- when using BP as the inference method (Table 2), the competing methods LC/Entropy are significantly slower than LinUProp
> 	- when using NETCONF as the inference method (Table 3), the runtime of LC/Entropy/CS/CS+LC is similar to that of LinUProp.
>
> - Further analysis of Tables 2 & 3
> 	- In Table 3, LinUProp's winners (BB/LC+BB) significantly outperform the winners of NETCONF-based UQ methods (CS/LC+CS) in all cases (pairwise t-test at a 5% significance level). Since NETCONF is never the winner among non-LinUProp methods, no significance is marked for it.
> 	- Beyond statistical significance, evaluating the stability of methods is also crucial. LinUProp achieves the highest mean accuracy in most cases compared to its competitors (28/32). Although not always significant, none of the competitors can maintain performance as good as LinUProp across all 4 datasets, 2 inference methods, and 4 labeling accuracies. Specifically, in Tables 2 & 3, while LinUProp isn't significantly better than LC/Entropy in some cases, it never exhibits high standard deviations (maximum standard deviation less than 7%) or very poor accuracy in any situation. In contrast, LC only achieves 55% accuracy on Polblogs in Table 3 across all four labeling accuracies, while LinUProp consistently exceeds 80%. The Entropy method shows high standard deviations on the Polblogs dataset in both tables (some exceeding 14%), whereas LinUProp generally has lower standard deviations compared to the non-LinUProp winners in the majority of all cases (marked with underlines).
>
> - Unique advantages of LinUProp compared to competitors
> 	-   LinUProp offers **interpretability** (Eq. (12)). For the posterior uncertainty of a node calculated by LinUProp, we can trace the contribution of any other node to that node's uncertainty, even if it's several hops away. This feature is not available in competing methods but is crucial for users to trust UQ results.
> 	-  LinUProp has a **solid theoretical foundation** (Section 4.2). We decompose the expected prediction error of the graphical model and prove that the uncertainty computed by LinUProp is the generalized variance component of the decomposition.

---

> > ### Comment · Reviewer_rPk6 · 2024-08-13
> >
> > Thank you. This is now a much more substantial, clear, and fair discussion of the relative performance and merits of LinUProp versus the other methods. As a reader, I feel I now have a better understanding of the contributions and performance of this method.
> >
> > Overall I'd like to thank the Authors for being responsive and providing the additional results and discussion that this paper needed. I feel it is now in a good place, and I will increase my score.

---

> ### Author Response · Authors · 2024-08-13
>
> Thank you for your valuable feedback, which has greatly improved our work. We're pleased that the expanded discussion and examples provide clarity, and we're glad the additional comparisons enhance the grounding of our results. Your insights on the performance and merits of LinUProp have helped create a clearer understanding. We appreciate your responsiveness and support in refining our paper.

---

### Official Review · Reviewer_XubP · 2024-07-12

**Soundness:** 3
**Presentation:** 4
**Contribution:** 3
**Rating:** 7
**Confidence:** 4

**Summary:**

This paper considers the problem of calculating uncertainty in the infererence results on probabilistic graphical models. Their method is based on a previously published linearization of the belief propagation method, to provide scalability, interpretability, and unbiasedness. The benefits of the new algorithm is demonstrated through experiments comparing with existing methods including Monte Carlo sampling, and uncertainty-based active learning on graphs.

**Strengths:**

Originality: the studied problem is unique in that a linearly scalable, interpretable, provably convergence uncertainty quantification methods on graph is still missing. The proposed method is novel with a new definition of uncertainty quantity that is based on interval width that is decomposed into bias and variance. Such a definition is not seen in prior work.
Quality: Technically, the formula and algorithm are developed rigorously and I don’t find errors therein. Their experiments protocols are designed carefully to evaluate their methods and prove the claimed advantages.
Clarity: the process of conducting the experiments, including datasets, graph constructions, baseline setup, are clearly described and reproducing the results is feasible. Figure 2 helps in understanding the dimension and operations of the linear operators on graphs.
Significance: their problem definition targets at gaps in uncertainty quantification on graphical model. In particular, previous methods, such as Monte Carlo and other uncertainty calculation methods on graphs, are either not scalable, biased, without convergence guarantee, or non-interpretable. In this aspect, the work contributes to the need of a scalable, interpretable, provably convergent, and unbiased uncertainty quantification algorithm for graphical models. In terms of theoretical significance, the decomposition of the computed interval width further makes sense of their proposed definition of uncertainty. Unlikely previous work that use variance of a distribution as a notion of uncertainty, the computed uncertainty is in fact a bias and variance term, thus shedding lights on the seemingly unmotivated definition of interval width.

**Weaknesses:**

-The proposed method is based on a previously developed linearization method, making their innovation limited.
-In the section of “Correctness of quantified uncertainty”, the correctness is only validated through a small simple graph, and no experiments are conducted on larger-scale graphs. Maybe the authors can explain why it is so designed?

**Questions:**

-In Figure 6, the legend shows 7 methods, while in some of the subfigures, there are only 5-6 curves. Can you explain why?
-In the same figure, LC seems to have performance between that of LC+BB and LC+CS. Is there any explanation about this observation?

**Limitations:**

Please see the weaknesses

---

> ### Author Rebuttal · Authors · 2024-08-07
>
> Thanks for your detailed and constructive reviews.
>
> >The proposed method is based on a previously developed linearization method, making their innovation limited.
>
> Indeed, we did use a conclusion from LinBP (Centered BP) in our derivation. However, LinUProp fundamentally differs from LinBP in its goals, and we have several unique contributions:
>
> -   **Different Goals**: The goal of LinBP is to infer the posterior beliefs of nodes, providing a point estimate of these beliefs. In contrast, LinUProp aims to quantify the uncertainty in the posterior beliefs, serving as a UQ method that offers the uncertainty in the form of interval widths.
> -   Since LinBP is not a UQ method, our contributions in the field of UQ are distinct from those of LinBP:
>     -   LinUProp addresses the issue of underestimating posterior uncertainty found in related works (e.g., NETCONF/SocNL).
>     -   The posterior uncertainty of nodes computed by LinUProp is interpretable, meaning we can understand the contribution of each other node to a specific node's posterior uncertainty by applying a Neumann series expansion to LinUProp.
>     -   We decompose the expected prediction error of the graphical model and prove that the uncertainty computed by LinUProp is the generalized variance component of the decomposition.
>
> >In the section of “Correctness of quantified uncertainty”, the correctness is only validated through a small simple graph, and no experiments are conducted on larger-scale graphs. Maybe the authors can explain why it is so designed?
>
> In the section “Correctness of Quantified Uncertainty,” we explained the reason for directly verifying the correctness of LinUProp on small graphs through MC simulations:
>
> "MC simulations are adopted as the ground-truth due to their ability to provide accurate approximations through a sufficient amount of sampling [30], which are feasible for small-scale graphs."
>
> For large-scale graphs, each sampling of the posterior belief for all nodes takes significantly longer than for small graphs, and more samples are needed to ensure accurate approximations. For example, on the Pubmed dataset, under the same experimental conditions as on small graphs, a single sampling of the posterior belief takes about 1 minute. Therefore, even maintaining the same number of samples as in the small graph experiment (100,000 times) would take approximately 70 days. Consequently, for large-scale graphs, we, like related works, validate the effectiveness of LinUProp in downstream tasks such as active learning (which we use)[1] , OOD detection [2], and robustness against node feature or edge shift [3].
>
> [1] Kang, Jian, et al. "JuryGCN: quantifying jackknife uncertainty on graph convolutional networks." SIGKDD, 2022.
>
> [2] Zhao, Xujiang, et al. "Uncertainty aware semi-supervised learning on graph data." NIPS, 2020.
>
> [3] Stadler, Maximilian, et al. "Graph posterior network: Bayesian predictive uncertainty for node classification." NIPS, 2021.
>
> >In Figure 6, the legend shows 7 methods, while in some of the subfigures, there are only 5-6 curves. Can you explain why?
>
> The curves corresponding to the BB and LC+BB methods are almost overlapping, which initially makes them appear as a single curve. However, upon closer inspection of the markers (such as $\triangle$ and $\triangledown$), it becomes evident that there are actually two distinct curves. The CS and LC+CS methods ($\triangleleft$ and $\triangleright$) exhibit a similar situation, ultimately making the 7 curves appear as only 5.
>
> >In the same figure, LC seems to have performance between that of LC+BB and LC+CS. Is there any explanation about this observation?
>
> The analysis of Figure 6 in the paper (lines 292-296) has already provided an explanation for this observation. This phenomenon nicely validates the potential issues with NETCONF that we mentioned in Figure 1: NETCONF (CS/LC+CS) prioritizes labeling low-degree nodes due to their inherent assumption that neighbors necessarily reduce uncertainty. As a result, nodes with many neighbors are often mistakenly viewed as having low uncertainty and are left unlabeled, which further leads to high uncertainty in the majority of nodes, ultimately causing LC+CS to perform worse than LC. In contrast, LinUProp (BB/LC+BB), which appropriately incorporates the uncertainty of neighbors, does not have this issue of underestimating uncertainty and thus results in LC+BB performing significantly better than LC.

---

### Official Review · Reviewer_HCKJ · 2024-07-12

**Soundness:** 3
**Presentation:** 3
**Contribution:** 3
**Rating:** 7
**Confidence:** 4

**Summary:**

A method for quantifying the uncertainty in the graphical model is proposed. The authors claimed that the method is superior over prior methods including NETCONF and Monte Carlo in the aspects of scalability, interpretability, and unbiasedness. Active learning that uses node uncertainty estimation for unlabeled node selection is conducted to show the benefits of the computed uncertainty.

**Strengths:**

- I find the decomposition of the prediction error into bias and generalized variance novel, and the assignment of the interval width to the variance term is creative.
- The submission is well-prepared and contain necessary components to make it a completed piece of work.
- The presentation of preliminaries clearly show the background to facilitate the understanding of the more complicated part in Section 3. Overall, the paper is well-organized with clear logics.
- The proof in Section 4 provide significant insight into the belief propagation algorithm and also the uncertainty quantification. Experiments on various setting proved the usefulness of the studied problem and the LinUProp method.

**Weaknesses:**

-	I have doubt about the interpretability of LinUProp, as it involves high-order terms such as $T^2$, $T^3$, etc. that are not easy to be understood.
-	Eq. (8) seems to be a simple extension of the original LinBP method.
-	It is not clear what are the $\mathbb{x}$, $\mathbb{y}$, and $\mathbb{P}$ terms in the proof in Eq. (9). What’s their relationship to the LinUProp method?

**Questions:**

-	The uncertainty is defined using interval width. Does the location of the interval matter and why?
-	Due to the proof of Eqs. (14) and (15), is it possible to compute the generalized variance component directly without LinUProp?

**Limitations:**

The clarity of Figure 2 can be improved: what is H_12, H_23, e_1, etc.? The authors should point to the definitions in the main texts or to label these symbols in the figure.

The GNN models are more popular, while its uncertainty quantification has been studied. Though GNN and probabilistic graphical models are not comparable, but I expect the authors to have certain discussion of GNN and make potential connections between methods for these two sorts of models.

---

> ### Author Rebuttal · Authors · 2024-08-07
>
> Thanks for your detailed and constructive reviews.
>
> >I have doubt about the interpretability of LinUProp, as it involves high-order terms such as $T^2$,$T^3$, etc. that are not easy to be understood.
>
> When we say that LinUProp is interpretable, we mean that for the posterior uncertainty of each node calculated by LinUProp, we can determine the contribution of each other node to the posterior uncertainty of that node. The matrices $T^2$,$T^3$, etc., are intermediate steps obtained from applying a Neumann series expansion to the closed-form solution of LinUProp. We do not need to understand their specific meanings; we only need to sum them to obtain the contribution of each other node to the current node's uncertainty. This allows us to achieve interpretability for LinUProp.
> Moreover, since the necessary and sufficient condition for LinUProp convergence is that the spectral radius of $T$ is less than 1, when the algorithm converges, the higher-order terms become very small. Therefore, in practice, it's often sufficient to compute only a few lower-order terms to understand the contribution of each other node to the uncertainty of a given node.
>
> >Eq. (8) seems to be a simple extension of the original LinBP method.
>
> Eq. (8) is the closed-form solution of LinUProp:
>
> $$\text{vec}(\mathbb{B})=(\mathbf{I}-\mathbf{\Psi}_{1}^{'}-\text{Diag}(\mathbf{\Psi_2}^{'}\mathbf{Q}))^{-1}\text{vec}(\mathbb{E}).$$
>
> The closed-form solution of LinBP is
>
> $$\text{vec}(\hat{\mathbf{B}})=(\mathbf{I}-\hat{\mathbf{H}}\otimes\mathbf{A}+\hat{\mathbf{H}}^2\otimes\mathbf{D})^{-1}\text{vec}(\hat{\mathbf{E}}).$$
>
> At first glance, Eq. (8) and LinBP appear somewhat similar, as both conform to the form $\mathbf{y}=(\mathbf{I-P})^{-1}\mathbf{x}$. However, simply replacing prior and posterior beliefs in LinBP with interval widths ($\text{vec}(\mathbb{E})$ and $\text{vec}(\mathbb{B})$) does not yield LinUProp. This is because LinUProp's goal is to calculate the uncertainty of each node's posterior belief, derived from the upper and lower bounds of messages and beliefs through a series of derivations, ultimately leading to Eq. (8). This cannot be achieved by merely extending LinBP; for a detailed derivation, please refer to Appendix A.1.
>
> >It is not clear what are the $\mathbb{x},\mathbb{y}$, and $\mathbb{P}$ terms in the proof in Eq. (9). What’s their relationship to the LinUProp method?
>
> After Eq. (9), on line 157, $\mathbf{y}=(\mathbf{I-P})^{-1}\mathbf{x}$ simply represents the general form of a linear equation system that can be solved using the Jacobi method. We merely want to convey that the closed-form solution of LinUProp conforms to this general form. Therefore, the convergence conditions are consistent with the Jacobi method. Specifically, $\mathbf{y}$ corresponds to $\text{vec}(\mathbb{B})$, $\mathbf{x}$ to $\text{vec}(\mathbb{E})$, $\mathbf{P}$ to $\mathbf{\Psi}_{1}^{'}+\text{Diag}(\mathbf{\Psi_2}^{'}\mathbf{Q})$.
>
> >The uncertainty is defined using interval width. Does the location of the interval matter and why?
>
> This can be concluded from the derivation of LinUProp. At the beginning of the derivation (Eqs. (16-19)), the upper and lower bounds of the messages and beliefs are involved. However, by Eq. (20), after subtracting the lower bound from the upper bound of the messages, only the interval width remains in the equation, and the specific location of the interval is no longer considered. Therefore, the specific location of the interval is not important for LinUProp.
>
> >Due to the proof of Eqs. (14) and (15), is it possible to compute the generalized variance component directly without LinUProp?
>
> Through Eqs. (14) and (15), we derived **the variance component** and further proved that the variance component is a special case of LinUProp. This ultimately led to the conclusion that LinUProp is the generalized variance component. It's important to note that "the generalized variance component" answers what the uncertainty computed by LinUProp represents; we cannot directly calculate "the generalized variance component" itself.
>
>
> >The clarity of Figure 2 can be improved: what is H_12, H_23, e_1, etc.? The authors should point to the definitions in the main texts or to label these symbols in the figure.
>
> We have stated in the caption of Figure 2 that the inputs to LinUProp include (1) Uncertainty in prior beliefs of each node represented as interval widths (2) Edge potentials. However, we did not explicitly specify in the caption how these symbols correspond to (1) and (2).
>
> In fact, $\mathbb{e}_1,\mathbb{e}_2,\mathbb{e}_3$ correspond to (1), and
>
> $\mathbf{H_{12}},\mathbf{H_{23}}$ correspond to (2). Although these symbols are defined in lines 123-128, explicitly mentioning this correspondence in the figure caption would make it clearer. We will explicitly include this correspondence in the caption of Figure 2 in the camera-ready version.
>
> >The GNN models are more popular, while its uncertainty quantification has been studied. Though GNN and probabilistic graphical models are not comparable, but I expect the authors to have certain discussion of GNN and make potential connections between methods for these two sorts of models.
>
> Some studies have attempted to combine GNNs with probabilistic graphical models to leverage the strengths of both approaches. For instance, this combination can provide interpretability to GNNs [1] or achieve performance that surpasses using GNNs alone in certain tasks [2]. Solely relying on UQ methods for GNNs may be insufficient to compute uncertainties for these hybrid approaches. Therefore, integrating LinUProp with existing UQ methods for GNNs could potentially be more promising.
>
> [1] Vu, Minh, and My T. Thai. "Pgm-explainer: Probabilistic graphical model explanations for graph neural networks." NIPS, 2020.
>
> [2] Qu, Meng, Yoshua Bengio, and Jian Tang. "Gmnn: Graph markov neural networks." ICML, 2019.

---

> > ### Comment · Reviewer_HCKJ · 2024-08-13
> >
> > Thank you for the detailed responses. I will keep my score after reviewing other referees' comments and the manuscript.

---

> > > ### Author Response · Authors · 2024-08-13
> > >
> > > Thank you for your feedback and for raising important points to improve our paper. We appreciate your insights.

---

### Author Rebuttal · Authors · 2024-08-07

Tables 2 and 3 have been updated with annotations for the standard deviation and t-test results.

---

### Decision · Program_Chairs · 2024-09-25

**Decision:**

Accept (poster)

**Comment:**

The reviewers agree that this is a solid, interesting and useful piece of work. There were some concerns about the presentation, the experimental setup and the novelty (relative to another piece of work, LinBP), but the authors have convincingly used their rebuttals to clarify these issues and suggest fairly minor changes to improve the paper.

To the authors: please look at the reviews and responses again, and please carefully take these into account when preparing the camera-ready version.